# REXTIME: A Benchmark Suite for Reasoning-Across-Time in Videos

**Jr-Jen Chen**[1]    **Yu-Chien Liao**[1]    **Hsi-Che Lin**[1]    **Yu-Chu Yu**[1]
**Yen-Chun Chen**[2]    **Yu-Chiang Frank Wang**[1]
[1]National Taiwan University    [2]Microsoft
rextime.github.io

## Abstract

We introduce REXTIME, a benchmark designed to rigorously test AI models' ability to perform temporal reasoning within video events. Specifically, REXTIME focuses on *reasoning across time*, *i.e.* human-like understanding when the question and its corresponding answer occur in different video segments. This form of reasoning, requiring advanced understanding of cause-and-effect relationships across video segments, poses significant challenges to even the frontier multimodal large language models. To facilitate this evaluation, we develop an automated pipeline for generating temporal reasoning question-answer pairs, significantly reducing the need for labor-intensive manual annotations. Our benchmark includes 921 carefully vetted validation samples and 2,143 test samples, each manually curated for accuracy and relevance. Evaluation results show that while frontier large language models outperform academic models, they still lag behind human performance by a significant 14.3% accuracy gap. Additionally, our pipeline creates a training dataset of 9,695 machine generated samples without manual effort, which empirical studies suggest can enhance the across-time reasoning via fine-tuning.

## 1   Introduction

Large Language Models (LLMs) and Multimodal Large Language Models (MLLMs) have nearly matched human performance in various language and vision-language tasks [1, 4, 35]. Notably, frontier MLLMs trained on web-scale proprietary datasets show impressive video understanding [2]. However, unlike LLMs which excel in text reasoning over long sequences, the cause-effect reasoning in MLLMs, especially in understanding long video events, remains under-explored. This capability is crucial in robotics and embodied agents [5, 29, 34], healthcare and medicine [19, 49], and law and policy making [19]. Despite the importance, current video-language tasks like moment retrieval [13, 20], highlights detection [20, 33], dense video captioning [7, 40], and video question answering [22, 37] mainly address text-visual alignment, overlooking deeper temporal reasoning challenges.

In an initial study, we identified a common shortcoming in the most advanced MLLMs – they struggle with video question answering when the question and answer correspond to different time segments. As shown in Fig. 1, the question *"How can we cut up the tomato efficiently?"* and the answer *"Hold up a plate and sharpen the knife on the plate."* each refer to separate segments. Surprisingly, a simple question like this can challenge leading MLLMs. Therefore, there is a pressing need for a benchmark to quantitatively assess video temporal reasoning. To address this, we introduce REXTIME, a benchmark to evaluate **Re**asoning-A**cross-Time** capabilities for video events.

To develop REXTIME, we propose an LLM-assisted data generation pipeline that minimizes human effort and cuts costs from $300 to $135 per 1,000 QA pairs. The benchmark includes **921** validation and **2143** test samples, each rigorously curated by human annotators. Empirical evidence indicates

38th Conference on Neural Information Processing Systems (NeurIPS 2024) Track on Datasets and Benchmarks.

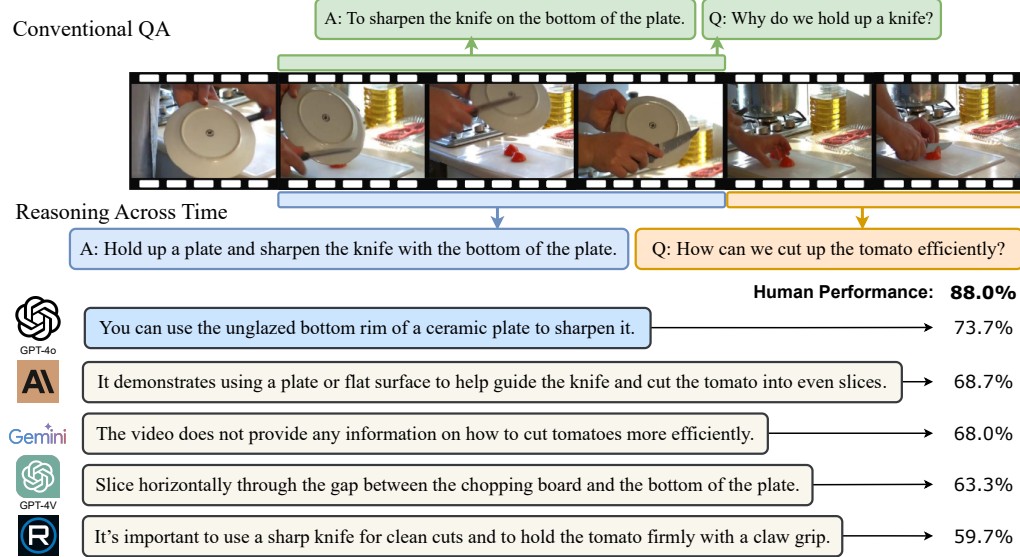

Figure 1: **A REXTIME example**. Our benchmark specializes in evaluating *reasoning across time*, *i.e.* video QA when question and answer each belong to different time spans. REXTIME poses difficulties even for frontier MLLMs, as indicated by the large gap to human-level accuracy.

that even proprietary frontier MLLMs are inadequate for temporal reasoning. For instance, humans can achieve 88.0% accuracy on VQA tasks, whereas the top-performing MLLM, OpenAI's GPT-4o, only reaches 73.7% as shown in Fig. 1. A new benchmark such as REXTIME has the potential to significantly propel advancements in this field – it effectively differentiates between model capabilities, and the state-of-the-art model has not yet saturated to human-level accuracy [30]. The additional 9695 unverified samples provide a training dataset that has significantly boosted an academic MLLM's temporal reasoning skills, lowering the entry bar for future research. Furthermore, we confirmed that REXTIME primarily contains *reasoning across time* questions, with the lowest question-answer overlap in time (QA-mIoU) compared to other video QA benchmarks.

To develop an efficient and effective pipeline, we have to address two primary challenges: (1) the quality-diversity trade-off in LLM generation, and (2) the high cost of human labor for verification. Initially, prompting an (M)LLM to generate question-answer pairs often results in logically incorrect responses. While few-shot in-context learning enhances logical correctness, it reduces response diversity. We address this by moderating the MLLM with specific event attributes and temporal relations from a structured taxonomy. Additionally, although human verification is necessary to eliminate residual errors, we minimize costs by establishing criteria that allow the MLLM to self-assess the accuracy of its generated QAs. As a bonus feature, we evaluate video moment localization to assess whether an AI model accurately grounds its answers to the correct video segments.

Our contributions can be summarized as the following:

- REXTIME is the first benchmark for comprehensive video temporal reasoning focusing on cause and effect with **2143** test samples, on which frontier MLLMs lag behind human performance.

- We discover a common weakness shared by current MLLMs – they reason poorly when question-answer spans do not overlap. We propose the measure of **QA-IoU**, which quantitatively validates that REXTIME better assesses AI models' ability in *reasoning across time*.

- Our LLM-assisted data pipeline generates high-quality samples with reduced human intervention, saving **55%** of the overall cost. Furthermore, the pure machine generated training set is shown to improve the finetuning accuracy, providing a starting point for future studies.

Table 1: **Dataset comparisons**. REXTIME covers features across different video QA tasks. Notably, *reasoning-across-time* emphasizes the cause-and-effect understanding between visual events.

| Datasets | QA | Moment Localization | Training Data | Temporal Reasoning | |
|---|---|---|---|---|---|
| | | | | sequential | causal |
| NExTQA [37] | ✓ | | ✓ | ✓ | |
| NExTGQA [38] | ✓ | ✓ | | ✓ | |
| Ego4D-NLQ [14] | | ✓ | ✓ | ✓ | |
| QVHighlights [20] | | ✓ | ✓ | | |
| REXTIME | ✓ | ✓ | ✓ | ✓ | ✓ |

## 2 Related work

**Temporal reasoning and event localization in videos**    In Table 1, we compare REXTIME with related datasets on temporal reasoning or moment localization, highlighting our uniqueness. NExTQA [37], enhancing video understanding by explaining temporal actions, specializes in temporal reasoning but not moment localization. NExTGQA [38], extends NExTQA with over 10.5K temporal grounding labels, revealing models' inadequacies in grounding answers despite strong QA performance. Ego4D-NLQ [14] lacks QA, making it difficult to assess modern AI chat assistants. QVHighlights [20] featuring over 10,000 YouTube videos across various themes, aiding systems in identifying relevant moments and highlights in response to user queries. However, it does not include temporal reasoning or QA pairs. Another related yet orthogonal work is EgoSchema [26], an extension of Ego4D, benchmarks long video comprehension and introduces the "certificate length" to measure intrinsic temporal complexity.

**Query depend moment retrieval**    Video moment retrieval involves retrieving specific video segments based on user text queries. Proposal-based methods [6, 9, 13, 15, 39, 46] use a two-stage process: generate candidate proposals by scanning the entire video and then rank them based on query alignment. In contrast, proposal-free methods [23, 42, 44] directly predict start and end timestamps or a center timestamp and span length. Recent approaches integrate the Detection Transformer (DETR) [8], leveraging its highlight detection capabilities [18, 20, 27, 28]. While these works focus on aligning visual and textual content, our research emphasizes temporal reasoning in scenarios with differing question and answer spans, requiring a distinct approach

**Grounding large video-language models**    In the evolving landscape of Multi-modal Large Language Models [4, 10, 24, 35, 41, 48], significant strides have been made in the realm of video understanding [21, 25, 43, 45, 47], particularly in the aspect of temporal localization [16, 17, 31, 32, 36]. VTimeLLM [16] excels with its boundary-aware training, improving Temporal Video Grounding and Dense Video Captioning. Momentor [31], using the Moment-10M dataset, enhances segment-level reasoning and localization, showcasing fine-grained temporal comprehension. HawkEye [36] focuses on complex videos with time-aware objectives and innovative segment representations, achieving notable performance gain in temporal video grounding. TimeChat [32] uses a timestamp-aware frame encoder and flexible video token generator for better long video understanding and zero-shot temporal reasoning. LITA [17] introduces time and SlowFast tokens [12], significantly improving temporal localization and video-text generation. These models collectively advance temporal understanding of multimodal AI. While they claim advanced temporal reasoning, there is no quantitative evaluation. To bridge this gap, we develop a comprehensive benchmark and dataset specifically designed to evaluate and enhance the temporal reasoning ability.

## 3 Data collection

We aim to collect video question-answer pairs to assess the *reasoning-across-time* capability of multimodal AI models. A conversation involves "reasoning-across-time" if the question's time span does not completely overlap with the answer's time span. By utilizing large language models and large vision language models, we create the benchmark, REXTIME, with much less human effort. Please refer to Fig. 2 for the data collection pipeline of REXTIME.

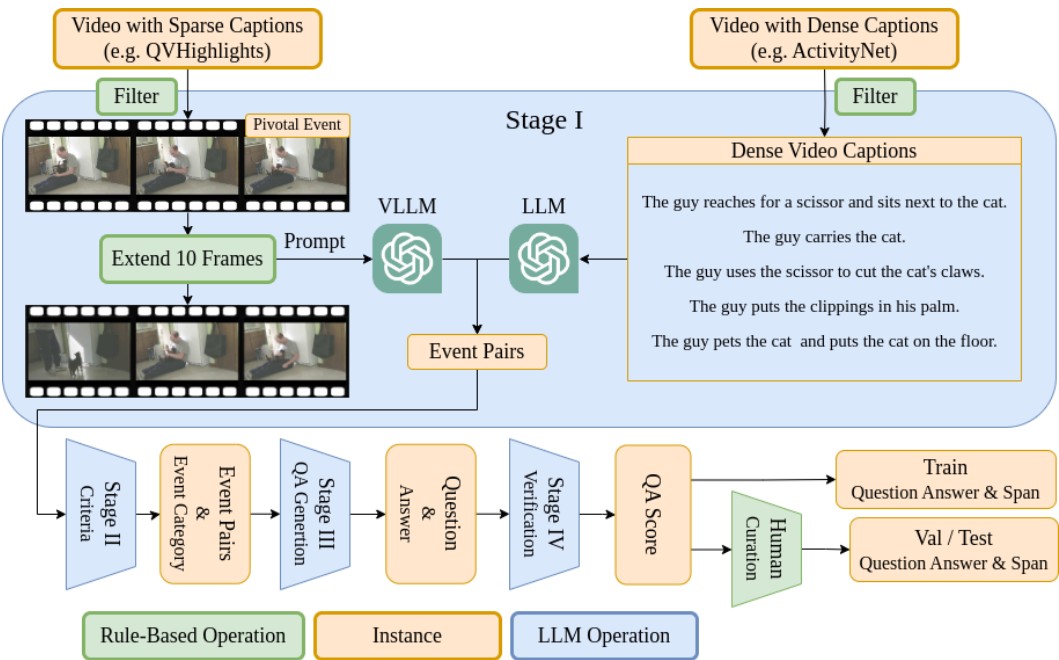

Figure 2: **Overview of the data collection pipeline.** In Stage I, we collect event pairs from two video sources ActivityNet [7] and QVHighlights [20]. In Stage II, we score and categorize the event pairs into three relation types: *Sequential*, *Cause-Effect* and *Means-to-an-End* (detailed in Sec. 3.2). In Stage III, the (M)LLM generates a question-answer pair by manually-defined few-shot demonstrations. To reduce the human verification cost, Stage IV utilizes the LLM to assess the reasonability of the generated samples.

## 3.1 Selecting videos to annotate

We consider video sources with time-aligned captions (*i.e.*, captions with start and end timestamps describing specific video segments) as they provide natural language descriptions of visual events crucial for video QA. We select ActivityNet [7] and QVHighlights [20] datasets, which meet this criterion, for QA data creation. To ensure the QAs focus on interesting events and involve reasoning across time, we apply rule-based filtering to retain only videos that: (1) contain at least two non-overlapping events, and (2) have events dense enough to cover the entire video duration. Further details on the filtering process are provided in the supplementary material.

## 3.2 Question-answering on two events across time

Naively feeding a video and its time-aligned captions to an MLLM often results in logically incorrect responses. Writing few-shot demonstrations improves correctness due to LLMs' strong in-context learning abilities but unexpectedly reduces diversity. To balance quality and diversity, grounding LLM generation in specific visual events and their relationships is essential. We extract event pairs from captions and categorize them into three relation types: *means-to-an-end*, *cause-effect*, and *sequential*. Means-to-an-end refers to one event causing another with subjective intentions, *i.e.*, "making a dish" leading to "chopping tomatoes." Cause-effect involves causal relations without a purpose, such as "girl falls down" causing "girl is crying." Sequential events are those with a "before / after" relation, where events do not completely overlap in time.

**Finding candidate event pairs** For QVHighlights videos, due to sparsely annotated captions (events), we use MLLM to find related events given an initial "pivotal event". We define a caption and its annotated time span as a "pivotal event" and crop the corresponding video clip with 10 second extensions before and after. We extend 5 predecessor and successor frames of the pivotal event, resulting in a total of 10 frames). For QVHighlights, since the frame rate is 2 seconds per frame, the above extension would indicate a 10-second extension in both directions (i.e., a total of

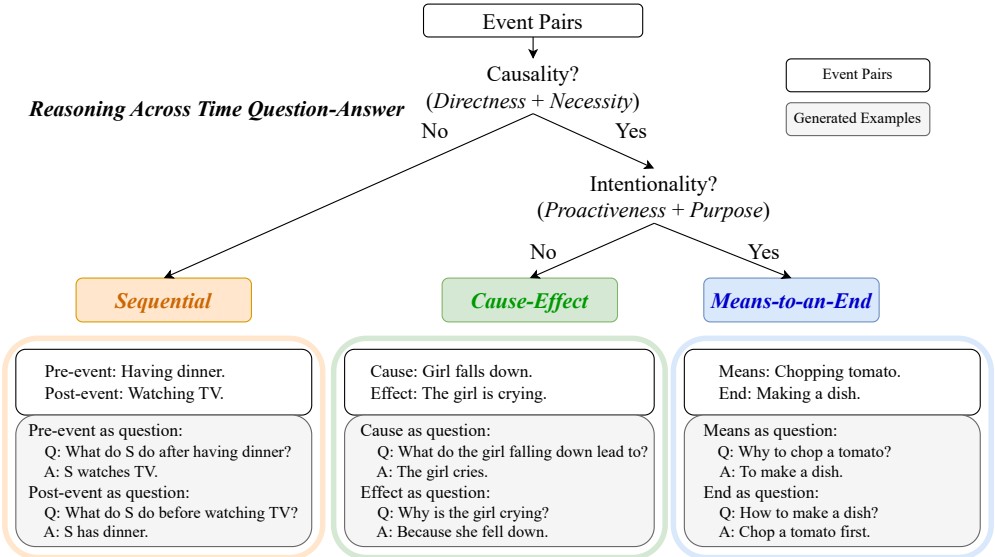

Figure 3: **Reasoning across time question-answer types** presents the relationship and examples between the three categories of question we generated. "Having dinner / Watching TV" does not have strong causality and is classified in *sequential*, which often results in before / after questions. "Girls falls down" shows strong causality with "The girl is crying." but lacks human intention, is classified in *cause-effect*. "Chopping tomato / Making a dish" not only has strong causal relations but also shows subjective deliberation, which is classified into *means-to-an-end*.

20 seconds). This extended clip is processed by GPT-4V to detect both the cause leading to the pivotal event and its consequent effects.

For ActivityNet videos, where events (captions) are denser, we use language-only GPT-4 to extract event pairs. We prompt the LLM to extract pairs with distinct timestamps and potential causal relations. These pairs are chosen based on their strong causal relationships, ensuring the events are temporally separated but intricately connected in terms of cause and effect.

To avoid selecting semantically identical events as candidate pairs, we ask the LLM to output a *similarity score* between events and only consider less similar pairs. For detailed prompts to GPT-4V and GPT-4, please see the supplementary material.

**Event relation classification**  We classify event pairs into the three aforementioned relations using the following four scoring criteria:

- *Directness*: This criterion assesses the directness of the causal link between events. For example, "A girl falls down. / She is crying." scores high in directness, while "A man has dinner. / He watches TV after dinner." scores low.

- *Necessity*: This criterion measures whether the second event is inevitable due to the first, *i.e.*, if the second event would still occur without the first. For example, "The marching band aligns in the street with instruments. / A man passes in front of the marching band holding a camera." scores high on *Directness*, but the second event is not necessarily a consequence of the first, resulting in a low *Necessity* score.

- *Proactiveness*: This evaluates whether an event is carried out with deliberate intention. Higher scores are given when there is clear evidence of premeditated action leading to the outcome. It can be viewed as proactiveness disregard of a successor event. For example, "Chop tomato. / Making a dish." scores high in *Proactiveness* because its human intention is clear.

- *Purpose*: Even if the preceding event is executed with intention, the resulting event may not align with the original expectation. We ask the LLM to specifically detect whether the intention has been fulfilled. For example, "Adding ingredients into a cup. / Putting a drink on the table."

scores high in *Proactiveness* but low in *Purpose* because the original goal was to make a drink, not to place it on a table.

We leverage GPT-4 to annotate these four scores $\in [0, 1, 2, 3]$ for each event pair. The relation can be classified using the following rules: (1) If the sum of directness and necessity scores is below $4$, they are in a simple *sequential* relation.[1] (2) If the sum of proactiveness and purpose is less than $5$, they are classified as a *cause-effect* relation. (3) If neither of the above conditions is met, the events are in a *means-to-an-end* relation. Figure 3 illustrates this process.

**Question-answer generation**   To generate QA pairs from the LLM, we crafted *in-context learning* [11] (ICL) examples specific to each event relation (see the ICL demonstrations in the supplementary material). To create a fair benchmark that can be automatically evaluated with reliable metrics, we made REXTIME a multiple-choice QA task. Thus, we need to generate negative options in addition to the ground truth answer. This is easily done with a language-only LLM, and the detailed prompt is provided in the supplementary material.

### 3.3   Balancing cheap machine generated data and high-quality human annotation

**Automatic data verification for cost reduction**   To ensure a high-quality benchmark, the correctness of the QA pairs is crucial, and a large sample size is needed to reduce variance in model evaluation. Therefore, we use LLMs to generate extensive data at a low cost, with human judges verifying the correctness of the output, which is faster than manual QA creation. To further reduce the rejection rate of LLM responses, we ask the LLM to self-verify the logical correctness of its outputs for cause-effect and means-to-an-end relationships (for sequential relations, the success rate is already high). Details of the prompts are provided in the supplementary materials. This step effectively reduces the human verification workload by filtering out poor samples. Due to the low access barrier of advanced LLMs, we generated more data than we could manually verify. Unverified data samples are used as the training dataset for REXTIME, serving as a jump-start dataset for future models to tackle our benchmark.

**Mitigating the modality misalignment**   A weakness of multiple-choice QA is that AI models can learn language-only shortcuts to achieve high accuracy. To address this, we require models to output the corresponding time span of the chosen answer. A stricter metric, accuracy with IoU @ $0.5$, may better reflect true multimodal understanding ability. One issue is that the annotated caption time spans from the original video corpus may not be accurate. Therefore, we request human annotators to re-annotate the event spans. The annotators are responsible for assessing each question-answer pair to ensure logical coherence and alignment with the video content, and for labeling the time span of the answer event.

## 4   Benchmark

### 4.1   Evaluation metrics

To evaluate performance, we use accuracy to assess multiple-choice VQA, where each question has four answer options. Additionally, we measure the model's ability to localize the answer event span using moment retrieval metrics, following Lei et al. [20]. We evaluate the Recall@1 score using two Intersection over Union (IoU) thresholds of $0.3$ and $0.5$. A model capable of multimodal understanding should excel in both VQA and localization, with accuracy @ IoU $\geq 0.5$ [38] being a key indicator.

### 4.2   How far are frontier MLLMs to solving REXTIME?

Table 2 shows the performance of humans and various multi-modal large language models, including GPT-4V [4], GPT-4o [2], Gemini [35], Claude [1], and Reka [3]. For evaluating MLLMs, we prompt the models to predict the time span directly and select the most likely options. We let the language models (such as GPT-4o) choose the correct answer from the four options, and see if the

---

[1]We further remove the pair if the two events are not consecutive to avoid answer ambiguity, *i.e.*, for "before / after" questions, we only consider the immediate preceding / following event.

Table 2: **Performances of human and frontier multi-modal large language models on the mini-test split (300 samples).** We randomly sampled 100 examples from each event relation category and evaluated API-based frontier MLLMs. Results show that while frontier MLLMs show certain degrees of temporal reasoning, they struggle with moment localization. We also estimate human-level performance, where each question is answered by three workers. The finding reveals that recent MLLMs are still far behind humans in both temporal reasoning VQA and moment localization.

| Models | Moment Localization | | | VQA | |
| --- | --- | --- | --- | --- | --- |
| | mIoU | R@1 (IoU=0.3) | R@1 (IoU=0.5) | Accuracy(%) | Accuracy(%) @IoU $\geq$ 0.5 |
| Human | **61.11** | **74.30** | **62.85** | **87.98** | **58.51** |
| GPT-4o [2] | **36.28** | **45.33** | **34.00** | **73.67** | **28.67** |
| Claude3-Opus [1] | 23.61 | 30.67 | 17.67 | 68.67 | 13.67 |
| Gemini-1.5-Pro [35] | 28.43 | 35.67 | 25.00 | 68.00 | 18.33 |
| GPT-4V [4] | 26.74 | 33.33 | 22.00 | 63.33 | 16.67 |
| Reka-Core [3] | 27.95 | 36.33 | 24.00 | 59.67 | 17.00 |

selected one matches the correct answer. Please refer to **??** in our supplementary for the detailed prompting process. Due to budget constraints and API query limits, we used a mini-test split of 300 samples. Human-level performance is included to set a benchmark for AI models and to identify future benchmark saturation.

In conclusion, the leading VLLMs can reason across time to some extent, as shown in the VQA accuracy. The newest MLLM, Reka, achieves $59.67\%$, and the best model, GPT-4o, achieves $73.67\%$. However, these models still lag behind the human-level accuracy of $87.98\%$. Despite claims of strong vision capabilities, these models often fail to localize the correct answer span, resulting in significantly lower mIoU compared to human performance.

### 4.3 Are academic and open source models competitive?

We consider both moment localization models [23, 27] and LLM-based models [16, 17, 32], and evaluate both zero-shot (Table 3) and fine-tuned performance (Table 4). A key observation is that most current open-source models struggle to accurately localize the ground truth moment in REX-TIME. Compared to proprietary frontier models, the zero-shot VQA accuracy of these open-source models is significantly lower. For pure VQA on temporal reasoning, humans can achieve $87.98\%$ accuracy, the best proprietary API achieves $73.67\%$, and the best open-source model only achieves $38.45\%$ accuracy. As contrasted, models trained on our dataset, as shown in Table 4, perform better on the moment retrieval task compared to the best proprietary API. The best-performing model, UniVTG, achieves an mIoU of $34.73\%$, which is competitive with frontier models at $36.28\%$. This indicates that frontier MLLMs are still not well-equipped for moment retrieval. Last but not least, we can see that after trained on our dataset, VTimeLLM gets a significant improvement from $36.25\%$ to $58.15\%$ on VQA. This result is even comparable to a frontier MLLM – Reka. Similarly, TimeChat improves from $38.45\%$ to $49.35\%$. Moreover, open source grounding language models can get a significant improvement on moment localization. In conclusion, utilizing our automatic generation pipeline, we can generate training data both effectively and efficiently with less than $10\%$ of the manual annotation cost in (see supplementary for detailed calculations). This could serve as a good starting point for future multimodal models' improvement on temporal reasoning.

### 4.4 Dataset statistics

**Question-answer intersection of union**    To quantify "across-time" reasoning, we introduce a new measure called Question-Answer Intersection over Union (QA-IoU). QA-IoU is calculated by dividing the intersection of the time spans of the question and answer by their union. A lower QA-IoU indicates a greater need for reasoning across time, as it reflects smaller time overlaps between the question and answer spans. To excel in a low QA-m(ean)IoU video question-answering task, a model must understand the temporal relationships between events, presenting significant challenges to modern multimodal AI assistants.

Table 3: **Zero-shot performance of open source models on the test split.** We assess the zero-shot capabilities of state-of-the-art moment retrieval models and grounding video LLMs. We choose two non-generative vision-language models [23, 27] and three LLM-based methods [16, 17, 32] with publicly available code and model weights. We can see open source models significantly lag behind frontier LLMs in temporal reasoning VQA.

| Models | Moment Localization | | | VQA |
|---|---|---|---|---|
| | mIoU | R@1 (IoU=0.3) | R@1 (IoU=0.5) | |
| UniVTG [23] | **28.17** | **41.34** | **26.88** | – |
| CG-DETR [27] | 23.87 | 31.31 | 16.67 | – |
| VTimeLLM [16] | 20.14 | 28.84 | 17.41 | 36.16 |
| TimeChat [32] | 11.65 | 14.42 | 7.61 | **40.04** |
| LITA [17] | 21.49 | 29.49 | 16.29 | 34.44 |

Table 4: **Test set performance of open source models after finetuning.** The results show that our fully automatic pipeline may provide useful training data to tech models to reason across time. We skip LITA [17] because the only publicly accessible model contains 13B parameters, which is beyond our computation resource to finetune.

| Models | Moment Localization | | | VQA | |
|---|---|---|---|---|---|
| | mIoU | R@1 (IoU=0.3) | R@1 (IoU=0.5) | Accuracy(%) | Accuracy(%) @ IoU $\geq$ 0.5 |
| UniVTG [23] | **34.63** | **53.48** | **34.53** | – | – |
| CG-DETR [27] | 26.53 | 39.71 | 22.73 | – | – |
| VTimeLLM [16] | 29.92 | 43.69 | 26.13 | **57.58** | **17.13** |
| TimeChat [32] | 26.29 | 40.13 | 21.42 | 49.46 | 10.92 |

**Average certificate lengths** Mangalam et al. [26] defined Certificate Length (C.L.) as the minimal length of the video segment necessary to answer a given question. In REXTIME, C.L. corresponds to the interval from the earliest start timestamp to the latest end timestamp of the question and answer spans. A longer Certificate Length requires the model to consider a longer segment to answer the question, increasing the difficulty for AI models.

**Comparison to similar tasks** Ego4D-NLQ is a task under the Ego4D Challenge [14] in the Episodic Memory category.[2] Given a video clip and a natural language query, Ego4D-NLQ requires a model to localize the temporal window within the entire video history where the answer to the question is evident. NExTGQA [38] extends NExT-QA [37] with 10.5k temporal grounding (or location) labels tied to the original QA pairs.

We compare REXTIME to the above two datasets on the number of reasoning across time samples, certificate length, and QA-mIoU. As depicted in Table 5, the average certificate length in our dataset is considerably longer than in existing tasks. This suggests that effectively addressing our task requires models to have more advanced temporal reasoning abilities.

The lower QA-mIoU in REXTIME indicates that an AI model needs to first locate the question event and then scan the rest of the visual events in the video to reason about the correct answer. This is more challenging because the reasoning and moment localization cannot be easily decomposed. For existing tasks, a model mostly needs to localize the question event and then reason within roughly the same span due to the higher QA-IoU.

Note that EgoSchema [26], which also poses significant challenges to modern deep learning systems, would be measured the longest certificate length mainly because its questions often ask for average

---
[2] https://ego4d-data.org/docs/challenge/.

Table 5: **Dataset statistics**. We focus on datasets with both question queries and moment localization features, and we list the number of temporal reasoning samples on each split, certificate length (C.L.) and Question-Answer mean Intersection over Union (QA-mIoU), respectively. A higher average certificate length indicates that the model needs to reason across a longer duration in a video. A lower QA-mIoU indicates smaller intersection of question span and answer span, requiring the model to reason across different time segments within a video. From the qualitative measures, REXTIME serves as a better benchmark to evaluate the model capability in reasoning across time. ([†]: Only counts temporal reasoning QA pairs. See supplementary for details.)

| Datasets | # of Reasoning Across Time Samples | | | C.L. (s) ↑ | QA-mIoU (%) ↓ |
|---|---|---|---|---|---|
| | Train | Val | Test | | |
| Ego4D-NLQ [14] | 2,212[†] | 775[†] | 705[†] | 5.2 | 85.5 |
| NExTGQA [38] | – | 1,403[†] | 2,301[†] | 11.7 | 66.1 |
| REXTIME | 9,695 | 921 | 2,143 | **66.0** | **15.5** |

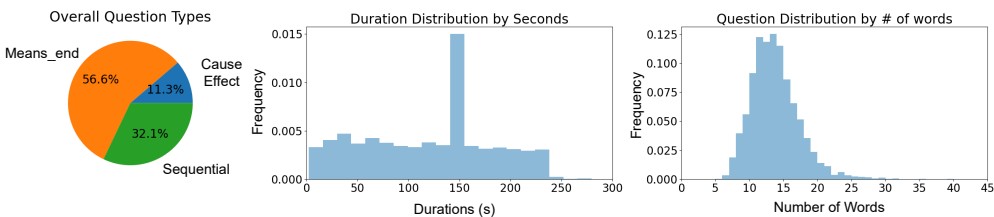

Figure 4: **Data distribution.** We visualize the distribution of the collected question-answer pairs. The pie chart shows the overall percentage of each relation category. The middle histogram shows the video duration distribution, while the right one depicts that of the number of words in a question. The lower number of *Cause-Effect* samples in ActivityNet can be attributed to the nature of the dataset, which predominantly features human activities. These activities typically involve deliberate actions with specific intentions, leading to a higher percentage of *Means-to-an-End* instances.

statistics or total counts of event occurrences throughout the video. Since this is not related to our focus on long-distance event relational reasoning, we do not include it in the table.

**Other statistics** Figure 4 provides additional analysis on question types, the distribution of question lengths in words, and the distribution of the total length of a video (video duration). We emphasize that REXTIME is diverse, as simple "before/after" questions account for less than 40% of the dataset, and a significant portion of the questions contain more than 15 words. Additionally, most videos are longer than 100 seconds, posing a challenging test for the multimodal model's ability to handle long sequences.

**BlindQA** To assess the QA quality, we choose to assess the BlindQA performance. In BlindQA, we feed only the language-based question and options without video content and ask the model to predict the answer. Using GPT-4o, the performances of VQA and BlindQA are 73.67% and 29.67%, respectively. Thus, it is clear that such an API model is not able to produce satisfactory performance in terms of BlindQA. This confirms the QA quality of our benchmark, suggesting that solving a temporal reasoning task is not trivial.

## 5 Conclusion

We proposed REXTIME, a comprehensive and reliable benchmark for multimodal AI, emphasizing *reasoning-across-time* and visual event localization in videos, with minimal human labor. We demonstrated that frontier MLLMs found REXTIME difficult and fall far behind human-level performance. The automatically constructed training dataset further points out a promising way for future models to equip the capability.

## Acknowledgement

We express our profound gratitude to all volunteers who contributed to the human annotation process for this project. The efforts have been essential in ensuring the data's accuracy and reliability, and our project's success could not have been realized without the commitment. Special thanks to Sheng-Yu Huang, I-Jieh Liu, Zi-Ting Chou, Bin Shih Wu, Yu Chi Pai, Yu-Chen Cho, Wei Wang, Yu Hsiang Huang, Hsin Chen Lin, Hung Kai Chung, Kuei Chun Wang, and Ko An Chu for your support and collaboration pivotal in achieving our project's goals.

This work is supported in part by the National Science and Technology Council via grant NSTC113-2640-E-002-003, the Featured Areas Research Center Program within the framework of the Higher Education Sprout Project by the Ministry of Education (MOE) of Taiwan via grant MOE113L900902, and the Center of Data Intelligence: Technologies, Applications, and Systems, National Taiwan University via grant NTU113L900902.

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
