# Supplementary Material
# REXTIME: A Benchmark Suite for Reasoning-Across-Time in Videos

**Jr-Jen Chen**[1]  **Yu-Chien Liao**[1]  **Hsi-Che Lin**[1]  **Yu-Chu Yu**[1]
**Yen-Chun Chen**[2]  **Yu-Chiang Frank Wang**[1]
[1]National Taiwan University   [2]Microsoft
[rextime.github.io](rextime.github.io)

# Appendix

38th Conference on Neural Information Processing Systems (NeurIPS 2024) Track on Datasets and Benchmarks.

## A  Changelog

### A.1  Version 2

- We have fixed some bugs in the evaluation code, resulting in slight differences compared to the previous release. The issue was that 149 samples were not evaluated in the previous version, and these have now been included in the new update.

### A.2  Version 3

- We have clarified certain statements and added experimental results to address the reviewer's questions.

## B  Additional documentation and resources

### B.1  Limitations

Despite these advancements, our dataset does exhibit certain limitations, largely stemming from inherited biases from the source datasets:

- Currently, we only address scenarios where both the question and the answer span a single time duration. Given a question, the annotated time span must be a single, continuous duration, which might be limiting for all scenes.

- The presence of noisy or inaccurate annotations in the source datasets, including captions and timestamps, poses a challenge. Despite our efforts, some of these errors could not be automatically filtered out. The extent of this issue is detailed in the qualitative visualization conducted by our human reviewers, as presented in supplementary.

- The average duration of ground truth events in our dataset is relatively long. This characteristic has the unintended consequence of hindering the models' ability to detect and analyze fine-grained actions within shorter video segments.

These drawbacks highlight areas for potential improvement and indicate the necessity for ongoing refinement to ensure the creation of more accurate and unbiased video language models.

### B.2  Social Impact

Though we provide an assessment of temporal reasoning and moment localization, the types and scene diversity are still limited. We inherit the video classes from the two source video datasets, which may not be sufficient for a comprehensive assessment of all kinds of temporal reasoning. This limitation could introduce a bias.

For both curated data and video data, they do not contain any personally identifiable information. Besides, some of the video samples in the source datasets might be slightly uncomfortable depending on the viewer. For example, some videos discuss tattoos and piercings, and some of them present news about social events including demonstrations or war reports. However, we only release the data of curated question-answer and time span. We are not responsible for the release and maintenance of video data.

### B.3 Data source links

Author's email: r12942106@ntu.edu.tw
Project page: https://rextime.github.io/
Huggingface dataset: https://huggingface.co/datasets/ReXTime/ReXTime
Github (code, data): https://github.com/ReXTime/ReXTime
Croissant: https://huggingface.co/api/datasets/ReXTime/ReXTime/croissant

### B.4 License

Our generated data is released under CC BY-NC-SA 4.0 license. Our code provided in Github is released under MIT license.

### B.5 Author statement

As the author of this work, we take full responsibility for any rights violations, including intellectual property rights. We confirm that all data used complies with applicable licenses and legal requirements, and all external sources have been properly credited and permissions obtained. This statement acknowledges our accountability and adherence to relevant data and copyright regulations.

### B.6 Maintenance plan

We will host and continuously update our data through various release sources, including GitHub (code and data), Huggingface (datasets), our project page, and the Eval.AI challenge.

### B.7 Digital object identifier (DOI)

You can find the digital object identifier in our citation block on Huggingface dataset page: https://huggingface.co/datasets/ReXTime/ReXTime

### B.8 Annotation instruction

We provide the link to the slide which is used in the annotation process as an instruction. Note that the used language in the slide is Chinese. Slide: https://docs.google.com/presentation/d/1-wgWYaWF-ZIa1YBSyPGc5p5TqTGXorxqkYZhOGIqqBg/edit?usp=sharing

## C  Additional implementation details

### C.1 Source Datasets

**ActivityNet**   ActivityNet is a comprehensive large-scale video benchmark designed to advance the field of human activity recognition by addressing the limitations of current computer vision algorithms. ActivityNet offers a diverse collection of complex human activities that reflect everyday life. The dataset encompasses 203 distinct activity classes, each with an average of 137 untrimmed videos, and features approximately 1.41 activity instances per video. This results in a substantial total of 849 video hours. ActivityNet supports various evaluation scenarios, including untrimmed video classification, trimmed activity classification, activity detection and dense video captions, making it a valuable resource for comparing and improving algorithms for human activity understanding.

**QVHighlights**   QVHighlights dataset addresses the challenge of detecting video moments and highlights based on natural language (NL) queries, an underexplored area due to a lack of annotated data. It includes over $10,000$ YouTube videos on various topics, each annotated with NL queries, relevant moments, and five-point saliency scores. This enables the development and evaluation of systems for detecting relevant moments and highlights. QVHighlights focused on user-created lifestyle vlog videos on YouTube. These videos, made by users worldwide, showcase various events and aspects of their lives, including everyday activities and travel. Captured with different devices (*i.e.*, smartphones, GoPro) and view angles (*i.e.*, first-person, third-person), they present significant challenges to computer vision systems. To enhance dataset diversity, we also included news videos

with substantial "raw footage", covering serious topics like natural disasters and protests. We used queries such as "daily vlog", "travel vlog", and "news hurricane" to harvest videos from YouTube, selecting top results between 5-30 minutes long, uploaded after 2016 for better visual quality, and filtering out videos with low view counts or high dislike ratios. These raw videos were then segmented into 150-second clips for annotation.

## C.2 Filter

The initial stage involves filtering out samples unsuitable for conversion into a temporal reasoning format. A temporal reasoning conversation sample requires a complex scene with sequential events occurring in it. Also, we need information which describes segments in detail instead of an overall summary of a whole video. Last but not the least, we want the sample source to contain as much information as possible. That's why we need a filter to select a proper sample source. For data originating from QVHighlights, we eliminate samples wherein the video content represents a single, continuous event. Specifically, this refers to videos where the answer span encompasses the entire duration, from start to finish. Also, we exclude samples if a query happens several times in the video, which indicates a routine and repeated behavior. In contrast, we apply a distinct set of criteria for filtering for samples from ActivityNet. First, samples with an event duration exceeding 80% of the total video length are discarded. This criterion helps ensure a diverse range of events within each video. Second, samples where the cumulative duration of all segments is less than 60% of the video's total length are regarded as insufficiently detailed ("sparse captioning") and are therefore excluded. This is due to potential information deficits in such samples. Third, we perform a clustering of event intervals, applying a threshold of 10 seconds. Intervals separated by gaps exceeding this threshold are considered discontinuous and are segmented into distinct groups. From these groups, we select the one with the highest event count for the generation of question-answer pairs, ensuring richness in temporal reasoning content.

## C.3 Cost estimation

**Test data generation and Verification**   We take 1000 samples as an example. One person can review 60 samples per hour. Generating 1000 samples with GPT-4 costs about 35$. At a minimum hourly rate of 6$, the total cost for 1000 samples, including human verification, is about 135$. Conversely, creating 20 natural language question-answer pairs for video content takes about one hour. Thus, generating 1000 samples would require 50 hours, costing 300$ in total. Our pipeline can create video QA data much more efficiently, at only 45% of the total cost.

**Training data generation**   We take 1000 samples as an example. Generating 1000 samples with GPT-4 costs about 35$. The total cost for generating 1000 training samples is about 35$. Conversely, creating 20 natural language question-answer pairs for video content takes about one hour. Thus, generating 1000 samples would require 50 hours, costing 300$ in total. Our pipeline can create reasoning-across-time training video QA data much more efficiently, at a bit more than 10% of the total cost.

## C.4 Computing resources

All of our fine-tuning experiments are done with an Nvidia RTX-3090 24G GPU.

## C.5 Training details and hyper-parameters

We report the training details and hyper-parameters in this section. Overall, we will follow the setting provided by the original papers or official Github setting. However, to fine-tune grounding video-language models such as [6, 11] on resource as reported in Appendix C.4, we will apply LoRA [5] fine-tuning and reduce batch size.

**UniVTG [8]**   We follow the single-gpu training script [1] provided by UniVTG official implementation with learning rate 1e-4, clip lengths 2, batch size 32, epochs 200 and hidden dimension 1024.

---

[1] https://github.com/showlab/UniVTG/blob/main/scripts/qvhl_pretrain.sh

We load the weight pre-trained on several datasets released by UniVTG official implementation for both zero-shot moment retrieval and fine-tuning experiments.

**CG-DETR [10]**   We load the weight pre-trained on QVHighlights released by CG-DETR official implementation for zero-shot moment retrieval. We follow the single-gpu training script [2] provided by CG-DETR official implementation to train on our generated data.

**VTimeLLM [6]**   To evaluate zero-shot performance, we load the stage 3 model weight from the VTimeLLM official implementation. We assess moment retrieval and VQA (Visual Question Answering) performance separately. For moment retrieval, we prompt the model with "Can you pinpoint when and..." followed by the question sentence, and extract the time token from the predicted sentence. For zero-shot VQA evaluations, we concatenate four options after the prefix "From <ss> to <ee>, <option>" as four predictions, here <ss> and <ee> is ground truth span. Then we calculate the sequence probability for each, and select the maximum probability as the VQA prediction.

For fine-tuning experiments, we follow the tuning strategy provided by VTimeLLM. Starting with the stage 3 model weight, we add a new LoRA adapter, tune on our generated training dataset, and merge the adapter during inference. We use the hyper-parameters from the original paper: a learning rate of 1e-4, number of video frames of 100, LoRA rank of 64, LoRA alpha of 128, training for 2 epochs, with a batch size of 8 and gradient accumulation steps of 16. For fine-tuned evaluation, we first predict the whole sentence given a question sentence and extract the predicted time tokens <ss> and <ee>. We then concatenate the four options after the predicted answer span "From <ss> to <ee>, <option>" as four predictions, calculate the sequence probability, and choose the maximum one for VQA and GQA (Grounding VQA) prediction. Here we provide a python pesudo as a demonstration:

---

**Pseudo code:**   This is a python pseudo code for the assessment of grounding multi-choice VQA.

```python
def extract_time_token(string):
    # string: From ss to ee, the girl is ....
    pattern = r"\s+(\d+)\s+to\s+(\d+)"
    matches = re.findall(pattern, string)
    return matches

def get_predicted_score(logits, labels):
    # Get label start index and end index
    start_idx, end_idx = ...
    scores = nn.CrossEntropyLoss(logits[start_idx:end_idx+1], \\
                                 labels[start_idx:end_idx+1])
    return scores

def concat(question, predicted_time_tokens, option)
    # question + 'From ss to ee' + option.
    return question + predicted_time_tokens + option

# Input: question(string), options(string in list, lenght==4)

# Time tokens prediction (Moment localization)
output = model.generate(question)
# Decode to natural language
response = tokenizer.decode(outputs)
# Extract time tokens
predicted_time_tokens = extract_time_token(response)

# Concatenate predicted_time_tokens with each option.
# From ss to ee, <Option>.
inputs = []
for i in range(4):
```

---

[2]https://github.com/wjun0830/CGDETR/blob/main/cg_detr/scripts/train.sh

```
    inputs.append(concat(question, predicted_time_tokens,
        options[i]))
inputs = tokenizer.encode(inputs)

# Multi-choice prediction (VQA)
# input_ids.shape==(4, batch_max_lenghts) for 4 options
output = model(**inputs['input_ids'])
# Compute the mean of labels sequence log-probability.
scores = get_predicted_score(output['logits'], inputs['labels'
    ])
# Find the one with largest crossentropyloss as predicted
    answer
predicted_answer = transition_scores.max()
```

**TimeChat [11]**    For the zero shot setting, we evaluate the checkpoints from the TimeChat official implementation. We also assess moment retrieval and VQA (Visual Question Answering) separately. For the first task we follow their prompt for temporal retrieval, and parse model's response to obtain the timestamps prediction. The evaluation process for zero-shot VQA for TimeChat is the same as that for VTimeLLM.

When fine-tuning TimeChat on our proposed dataset, we start from fine-tuned checkpoints provided by TimeChat and follow their instruction fine-tuning settings. Specifically, we use LoRA with a rank of 32, alpha of 128. We train the model with a learning rate of 3e-5, batch size of 8, and gradient accumulation steps of 8 for 3 epochs. The number of frames used in each video is 96. To evaluate the performance after fine-tuning, we use the same evaluation protocol as that we used for VTimeLLM.

### C.6    Counting temporal reasoning QAs

We compare REXTIME with Ego4D-NLQ and NExTGQA. For metrics like **average certificate lengths (C.L.)** and **question-answer intersection of union (QA-IoU)**, we follow the methodology from Mangalam et al. [9], manually annotating at least two hours of human effort for each dataset. A screenshot of the labeling GUI tool is provided in Appendix C.7.

To determine the **number of reasoning across time samples**, we count the total queries with "be-fore/after" in Ego4D-NLQ and the samples of the "temporal" type in NExTGQA. We exclude other cases where the question time span completely overlaps with the answer time span, as they do not qualify as "reasoning across time."

### C.7    GUI

To facilitate efficient annotation, we have developed a Gradio graphical user interface (GUI).[3] Here we provide three types of annotation tool, human time span annotation and verification, human question span annotation and human performance annotation. Here we show in Fig. 1. The first one is for human time span annotation. The annotators are responsible for assessing each question-answer pair to ensure logical coherence and alignment to the video content. Additionally, they need to provide the time span of the answer, which will be used as ground truth in the following. The second one is for human question span annotation, given question, answer and answer span, the annotators need to find a span which is relevant to the question event. This is for the assessment of average certificate lengths (C.L.) and question-answer intersection of union (QA-IoU). The third one is for human performance experiment, given question and four options, the participants need to find not only the answer from the four options but also an answer span which is relevant to the selected answer. The green area indicates what an annotator will get, and the orange area indicates what an annotator need to answer.

### C.8    Prompts

### C.8.1    ActivityNet event generation

---

[3] https://www.gradio.app/

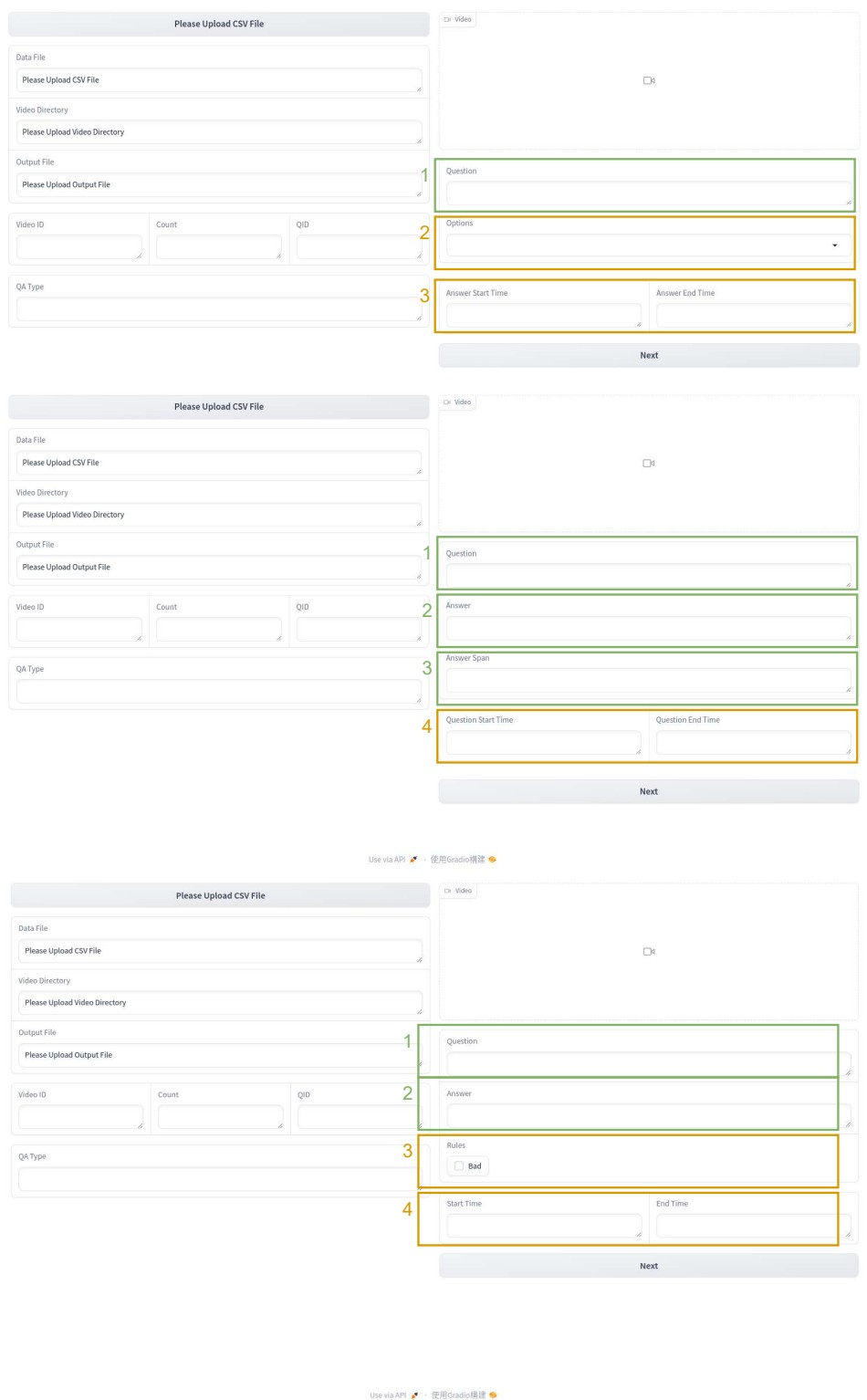

Figure 1: We show the GUIs for different annotation / verification processes.

```
Following the steps below to evaluate the causality between two
    events in the video.
First, find two events from different timestamps which have
    strong causality.
Second, evaluate the causality between the two events according
    to the following criteria:
a. Directness: How directly does one event lead to the next?
b. Necessity: Is the subsequent event a necessary consequence
    of the previous one?
c. Proactiveness: Determine if the first event is deliberately
    executed to cause the second event.
d. Purpose: Assess whether the first event is conducted with
    the primary goal of leading to the second event.
Scoring Method for criteria a to d (Score 0-3 for each
    criterion):
0: Weak causal relationship.
1: Moderate causal relationship.
2: Strong causal relationship.
3. Definite causal relationship.

e. Similarity: Assess whether the two events are just repeated
    actions or not.
Scoring Method for criterion e (Score 0-3):
0: Totally different action.
1: Slightly same action with event progression.
2: Partially same action with little event progression.
3: Totally same action without event progression.

Sequential video captioning:
<CAPTIONS>
Provide a brief and concise explanation of the score you give
    to each criterion.
<Provide your explanation here>
Finish the result json according to your evaluation:
```json{
    "event1": "<EVENT1>",
    "event1_timestamp": [start, end],
    "event2": "<EVENT2>",
    "event2_timestamp": [start, end],
    "Directness": <DIRECTNESS>,
    "Necessity": <NECESSITY>,
    "Intentionality": <INTENTIONALITY>,
    "Purpose": <PURPOSE>,
    "Similarity": <SIMILARITY>
}```
```

### C.8.2 QVHighlights event generation

```
These are frames from a video.
Find out a behavior in the video which is caused by the pivotal
    event "<QUERY>" or a behavior which leads to the pivotal
    event."
If there isn't any behavior that is caused by or leads to the
    pivotal event, return "none".
Your response should be in json format as the following.
{
    "explain": <A brief explanation according to the video and
        instruction>,
```

```
      "cause": <The behavior leads to the pivotal event>,
      "cause-relevant": <Does the cause have strong temporal-
          causality with the pivotal event? yes or no.>,
      "cause-alignment": <How well the cause is aligned with the
          video? high, medium, low>,
      "effect": <The behavior caused by the pivotal event>,
      "effect-relevant": <Does the effect have strong temporal-
          causality with the pivotal event? yes or no.>,
      "effect-alignment": <How well the effect is aligned with
          the video? high, medium, low>
}
```

### C.8.3 Sequential QA generation

```
Sequential video captioning:
<CAPTIONS>

Find two continuous events in the video captions from different
    timestamps.
Construct a temporal related question and answer based on the
    two events.
Examples:
(Pre-event) Jack wakes up. (Post-event) Jack brushes his teeth.
Type1. Question (pre-event): What does Jack do after waking up?
    Answer (post-event): Jack brushes his teeth.
Type2. Question (post-event): What does Jack do before brushing
    his teeth? Answer (pre-event): Jack wakes up.

Provide a brief and concise explanation.
<Your brief explanation here>
Finish the result json according to your explanation:
```json{
    "pre-event": "<EVENT1>",
    "pre-event_timestamp": [start, end],
    "post-event": "<EVENT2>",
    "post-event_timestamp": [start, end],
    "Type1": {
        "Question": "<QUESTION>",
        "Answer": "<ANSWER>"
    },
    "Type2": {
        "Question": "<QUESTION>",
        "Answer": "<ANSWER>"
    }
}```
```

### C.8.4 Cause-effect QA generation

```
This is a cause-effect relationship. The event "<EVENT1>"
    causes the event "<EVENT2>".
Please construct 2 types of questions and answers based on the
    cause-effect relationship.
Examples:
(Cause) A girl falls off a bike. (Effect) She is injured.
Type1. Question (cause): What does the girl falling off the
    bike lead to? Answer (effect): She is injured.
Type2. Question (effect): Why is the girl injured? Answer (
    cause): She falls off the bike.
```

```
Provide a brief and concise explanation.
<Your brief explanation here>
Finish the result json according to your explanation:
```json{
    "Type1": {
        "Question": "<QUESTION>",
        "Answer": "<ANSWER>"
    },
    "Type2": {
        "Question": "<QUESTION>",
        "Answer": "<ANSWER>"
    }
}```
```

### C.8.5 Means-to-an-end QA generation

```
This is a means-to-an-end relationship. The event "<EVENT1>" is
    a means to achieve the event "<EVENT2>".
Please construct a question and an answer based on the means-to
    -an-end relationship.
Examples:
(Means) Mixing flour and water. (End) Make dough.
Type1. Question (end): How do we make dough? Answer (means): By
    mixing flour and water.
Type2. Question (means): Why do we mix flour and water? Answer
    (end): To make dough.

Provide a brief and concise explanation.
<Your brief explanation here>
Finish the result json according to your explanation:
```json{
    "Type1": {
        "Question": "<QUESTION>",
        "Answer": "<ANSWER>"
    },
    "Type2": {
        "Question": "<QUESTION>",
        "Answer": "<ANSWER>"
    }
}```
```

### C.8.6 QA verification

```
You are tasked with verifying if the question-answer pair is
    logically correct.
Provide your explanation of the verification result.
Provide a score from 0 to 3 to indicate the correctness of the
    question-answer pair.
Scoring Method:
0: Incorrect question-answer pair.
1: Relevant but not logically correct.
2: Partially correct.
3: Completely correct.

Case 1:
Question: <QUESTION1>
Answer: <ANSWER1>
```

```
Case 2:
Question: <QUESTION2>
Answer: <ANSWER2>

Provide a brief and concise explanation.
<Your brief explanation here>
Finish the result json according to your explanation:
```json{
    "case_1_score": <SCORE>,
    "case_2_score": <SCORE>
}```
```

### C.8.7 Options generation

```
You are tasked with generating high-quality, incorrect options
    for a given question-answer pair.
The options should be logically consistent and correct, but
    they must be different from the correct answer.
Note that the generated options should not be a interpretations
     or variations of the correct answer in any way.

Here are the question-answer pair:
Question: <QUESTION>
Answer: <ANSWER>

Provide a brief and concise explanation.
<Your brief explanation here>
Finish the result json according to your explanation:
```json{
    "correct_answer": "<ANSWER>",
    "wrong_answer_1": "<OPTION1>",
    "wrong_answer_2": "<OPTION2>",
    "wrong_answer_3": "<OPTION3>"
}```
```

## D  Additional experiment results

### D.1  Qualitative Results

We present six generated data examples in Fig. 2, categorized into three types: *sequential*, *cause-effect*, and *means-to-an-end*. For each category, we include one example where the question event happens first and another where the answer event happens first.

### D.2  Teaser examples

Here we provide the inference case depicted in teaser on GPT-4V Achiam et al. [3], Gemini-1.5-Pro Team et al. [12], Claude3-Opus Clu [1] and Reka-Core Rek [2].

### D.2.1  GPT-4V

**Number of Frames:** 50
**Prompt Input:**

```
According to the video, answer the following questions:
How can we cut the tomato more efficiently?
```

**Response:**

Q: What does the girl do before she begins walking backwards doing a flip?

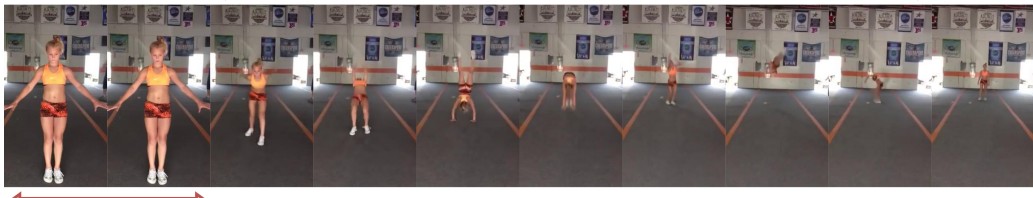

A: A girl is seen standing on a mat looking down at the floor.

Q: What happens after the two fall to the floor at the end of the slide?

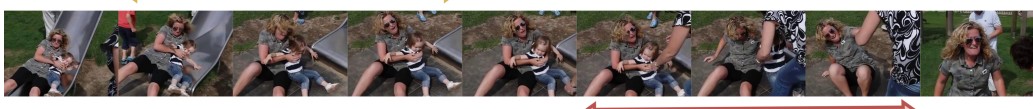

A: Someone assists the two in getting up from the ground.

Q: How does the boy ride down the slide on the skateboard?

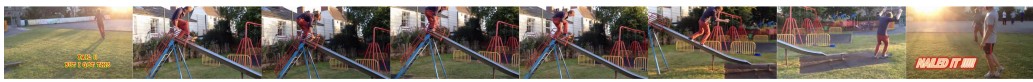

A: By positioning and standing on the skateboard.

Q: What happens after the woman explains and gets off?

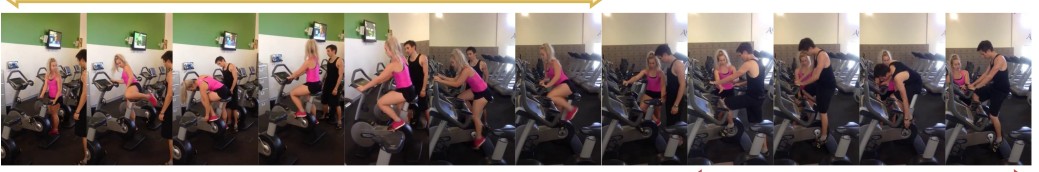

A: The man gets on and uses the machine as the girl explains further.

Q: Why does the dog get water everywhere?

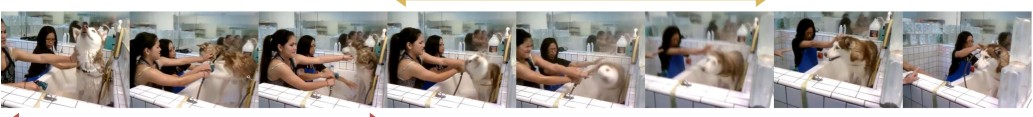

A: Because the dog does not like the bath and struggles.

Q: Why does he bend down, lift her up to his face, and flip her around?

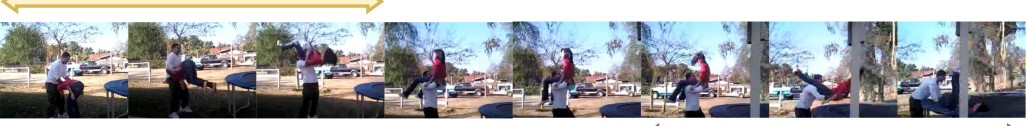

A: To slam the woman down on her back onto a trampoline.

Figure 2: We show the qualitative results of generated data.

```
To cut the tomato more efficiently, the video demonstrates a
   technique where you can use a plate as a guide to slice the
   tomato. Here are the steps based on the images provided:

   1. Place the tomato on a chopping board.
```

```
          2. Cover the tomato with a flat, slightly deep plate,
             upside down.
          3. Hold the plate firmly in place over the tomato.
          4. Using a sharp knife, slice horizontally through the gap
             between the chopping board and the bottom of the plate.
              This allows you to slice the tomato into even sections
              while keeping your fingers safely away from the knife.
          5. Carefully lift the plate to reveal the evenly sliced
             tomato segments beneath.

This method is particularly useful for slicing multiple
   tomatoes quickly and uniformly, and it also helps in
   keeping the slices contained, making it cleaner and more
   organized.
```

### D.2.2 Gemini-1.5-Pro

**Number of Frames:** 50
**Prompt Input:**

```
According to the video, please describe the video content.
```

**Response:**

```
A man is in a kitchen. He is preparing a meal. He is cutting
   tomatoes and there are other ingredients on the counter,
   such as meat, green beans, and oil. The man is wearing a
   hat and a light blue shirt.
```

**Prompt Input:**

```
According to the video, answer the following questions: How can
    we cut the tomato more efficiently?
```

**Response:**

```
The video does not provide any information on how to cut
   tomatoes more efficiently. The video only shows a person
   cutting tomatoes.
```

### D.2.3 Claude3-Opus

**Number of Frames:** 20
**Prompt Input:**

```
According to the video, answer the following questions: How can
    we cut the tomato more efficiently?
```

**Response:**

```
The video demonstrates using a plate or flat surface to help
   guide the knife and cut the tomato into even slices more
   efficiently. By holding the tomato against the plate while
   slicing downward with the knife, it allows for making
   consistent, thin slices quickly without the tomato rolling
   around on the cutting board. The plate acts as a stable
   guide to help control the slicing motion and produce
   uniform tomato slices with less effort compared to trying
   to slice the tomato freehand.
```

### D.2.4 Reka-Core

**Number of Frames:** Unknown
**Prompt Input:**

```
According to the video, answer the following questions: How can
    we cut the tomato more efficiently?
```

**Response:**

```
The video demonstrates a method for slicing a tomato. To cut
    the tomato more efficiently, you should start by slicing
    off the bottom of the tomato to create a stable base. Then,
    slice the tomato horizontally into even rounds, depending
    on your preference. If you want to dice the tomato, after
    slicing it into rounds, stack the slices and cut across
    them to create cubes. It is important to use a sharp knife
    for clean cuts and to hold the tomato firmly with a claw
    grip to avoid cutting your fingers.
```

**Prompt Input:**

```
According to the video, answer the following questions: Why do
    we hold up a plate?
```

**Response:**

```
In the video, the act of holding up a plate can have several
    purposes. It could be to inspect the plate for cleanliness,
    to show the plate to someone else, or to demonstrate a
    technique or feature of the plate, such as a logo or a
    special design on the bottom. It could also be part of a
    cooking demonstration or instructional video where the
    person is explaining the next step or ingredient to the
    audience.
```

### D.2.5 GPT-4o

**Number of Frames:** 50
**Prompt Input:**

```
According to the video, answer the following questions: How can
    we cut the tomato more efficiently?
```

**Response:**

```
Based on the sequence of images from the video, here is how you
    can cut a tomato more efficiently:
**Use a Sharp Knife: Ensure your knife is sharp. The video
    shows the person demonstrating the sharpness of the knife
    using the edge of a plate.
**Sharpen the Knife**: If your knife is not sharp enough, you
    can use the unglazed bottom rim of a ceramic plate to
    sharpen it. Run the knife's edge along the rim at an angle
    to hone the blade. This can be a handy trick if you don't
    have a knife sharpener.
**Proper Technique**:
    1. Start by halving the tomato.
    2. Place the tomato halves on the cutting board with the
        flat side down for stability.
```

3. Slice the tomato into even pieces, keeping your fingers
            tucked safely away from the blade.
By ensuring your knife is sharp and using effective slicing
   techniques, you can cut the tomato more efficiently and
   safely.

Table 1: Zero-shot performance on mini test set. We report open source models' zero-shot performance on the mini test set compared to the full set.

| Models | Moment Localization | | | VQA | |
| --- | --- | --- | --- | --- | --- |
| | mIoU | R@1 (IoU=0.3) | R@1 (IoU=0.5) | Accuracy(%) | Accuracy(%) @ IoU $\geq$ 0.5 |
| UniVTG (zs) | 28.17 | 41.34 | 26.88 | – | – |
| UniVTG (zs,mini) | 30.18 | 42.00 | 29.33 | – | – |
| CG-DETR (zs) | 23.87 | 31.31 | 19.60 | – | – |
| CG-DETR (zs,mini) | 22.53 | 30.00 | 16.67 | – | – |
| VTimeLLM (zs) | 20.14 | 28.84 | 17.41 | 36.16 | – |
| VTimeLLM (zs,mini) | 19.37 | 27.67 | 16.00 | 37.33 | – |
| TimeChat (zs) | 11.65 | 14.42 | 7.61 | 40.04 | – |
| TimeChat (zs,mini) | 13.01 | 16.33 | 7.00 | 38.33 | – |
| LITA (zs) | 21.49 | 29.49 | 16.29 | 34.44 | – |
| LITA (zs,mini) | 24.76 | 34.33 | 20.00 | 35.00 | – |

Table 2: Fine-tuned performance on mini test set. We report open source models' fine-tuned performance on the mini test set compared to the full set.

| Models | Moment Localization | | | VQA | |
| --- | --- | --- | --- | --- | --- |
| | mIoU | R@1 (IoU=0.3) | R@1 (IoU=0.5) | Accuracy(%) | Accuracy(%) @ IoU $\geq$ 0.5 |
| UniVTG (ft) | 34.63 | 53.48 | 34.53 | – | – |
| UniVTG (ft,mini) | 34.82 | 53.00 | 35.33 | – | – |
| CG-DETR (ft) | 26.53 | 39.71 | 22.73 | – | – |
| CG-DETR (ft,mini) | 24.98 | 38.00 | 20.33 | – | – |
| VTimeLLM (ft) | 29.92 | 43.96 | 26.13 | 57.58 | 17.13 |
| VTimeLLM (ft,mini) | 29.53 | 43.67 | 25.00 | 54.67 | 15.00 |
| TimeChat (ft) | 26.29 | 40.13 | 21.42 | 49.46 | 10.92 |
| TimeChat (ft,mini) | 27.54 | 38.00 | 21.67 | 52.00 | 11.33 |

### D.3 Open source performance on mini test set

We show the performance results of open source models on the mini test set. Please refer to Table 1 and Table 2.

# E Datasheet for REXTIME

**For what purpose was the dataset created?** Was there a specific task in mind? Was there a specific gap that needed to be filled? Please provide a description.

REXTIME is a benchmark designed to rigorously test AI models' ability to perform temporal reasoning within video events. Specifically, REXTIME focuses on *reasoning across time*, *i.e.* human-like understanding when the question and its corresponding answer occur in different video segments. This form of reasoning, requiring advanced understanding of cause-and-effect relationships across video segments, poses significant challenges to even the frontier multimodal large language models. To facilitate this evaluation, we develop an automated pipeline for generating temporal reasoning question-answer pairs, significantly reducing the need for labor-intensive manual annotations. Our benchmark includes 921 carefully vetted validation samples and 2,143 test samples, each manually curated for accuracy and relevance. Evaluation results show that while frontier large language models outperform academic models, they still lag behind human performance by a significant 14.3% accuracy gap. Additionally, our pipeline creates a training dataset of 9,695 machine generated samples without manual effort, which empirical studies suggest can enhance the across-time reasoning via fine-tuning.

**Who created this dataset (e.g., which team, research group) and on behalf of which entity (e.g., company, institution, organization)?**

The authors created the dataset at National Taiwan University, Graduate Institute of Communication Engineering, Vision and Learning Laboratory.

**Who funded the creation of the dataset?** If there is an associated grant, please provide the name of the grantor and the grant name and number.

National Taiwan University.

**Any other comments?**

**What do the instances that comprise the dataset represent (e.g., documents, photos, people, countries)?** Are there multiple types of instances (e.g., movies, users, and ratings; people and interactions between them; nodes and edges)? Please provide a description.

Each instance in the dataset includes a question-answer pair along with the corresponding video ID. Each answer corresponds to a specific time span in the video. Additionally, we have collected four options for the validation and test sets to facilitate evaluation.

**How many instances are there in total (of each type, if appropriate)?**

There are 9,695 training samples, 921 validation samples, 2,143 test samples and 300 samples in mini test set. In 9,695 training samples, there are 1,469 *cause-effect* samples, 4804 *means-to-an-end* samples, and 3422 *sequential* samples. In 921 training samples, there are 74 *cause-effect* samples, 515 *means-to-an-end* samples, and 332 *sequential* samples. In 2,143 training samples, there are 247 *cause-effect* samples, 1089 *means-to-an-end* samples, and 807 *sequential* samples. In 300 training samples, there are 100 *cause-effect* samples, 100 *means-to-an-end* samples, and 100 *sequential* samples.

**Does the dataset contain all possible instances or is it a sample (not necessarily random) of instances from a larger set?** If the dataset is a sample, then what is the larger set? Is the sample representative of the larger set (e.g., geographic coverage)? If so, please describe how this representativeness was validated/verified. If it is not representative of the larger set, please describe why not (e.g., to cover a more diverse range of instances, because instances were withheld or unavailable).

This dataset is collected from two moment localization datasets including ActivityNet [4] and QVHighlights [7]. In this work, we employed two different strategies, as shown in Section 3.1, to filter data samples from the source datasets based on their intrinsic features. Additionally, we filtered out certain generated question-answer pairs according to a designed criterion and scored by GPT-4, as depicted in paper. We excluded some data samples because they were unsuitable for conversion to temporal reasoning cases, and some generated samples did not constitute reasonable question-answer pairs.

**What data does each instance consist of? Raw data (e.g., unprocessed text or images) or features?** In either case, please provide a description.

Each instance in the raw data includes a question ID, a corresponding video ID, a question, video total duration in seconds, meta data with question type(sequential, cause-effect, means-to-an-end), video source(ActivityNet, QVHighlights). Samples in training set contain an answer sentence and relevant answer time span. Samples in validation set contain four options options, answer option and relevant answer time span. Samples in test set contain only four options without answer and answer span, which are hidden for private evaluation on our benchmark. We do not release the mini test set which all of them are in the original test set.

**Is there a label or target associated with each instance?** If so, please provide a description.

A label can be the answer sentence in the conversation for generated task. Also, for multi-choice, we provide the correct option in 'a', 'b', 'c' or 'd'. Each answer sentence is corresponding to a relevant time span.

**Is any information missing from individual instances?** If so, please provide a description, explaining why this information is missing (e.g., because it was unavailable). This does not include intentionally removed information, but might include, e.g., redacted text.

We do not provide the corresponding time spans for question sentences in our dataset due to the difficulty of collecting this information.

**Are relationships between individual instances made explicit (e.g., users movie ratings, social network links)?** If so, please describe how these relationships are made explicit.

Some instances may have the same video but different questions and answers. It can be distinguished by their unique question ID.

**Are there recommended data splits (e.g., training, development/validation, testing)?** If so, please provide a description of these splits, explaining the rationale behind them.

We provide generated training data without human verification for efficient collection and scalable training. Our validation and test sets are automatically collected and annotated by humans. This setup allows for zero-shot assessment of multi-modal large language models using a private API. It also allows for fine-tuned assessment of open source models.

**Are there any errors, sources of noise, or redundancies in the dataset?** If so, please provide a description.

Yes, we have carefully designed the pipeline to convert dense video caption data or moment retrieval data into reasoning across time question-answer pairs. However, some of the generated samples are still not reasonable for humans and cannot be filtered out. Additionally, the answer time spans collected from the training data inherit noise from the source dataset, leading to inconsistencies with human preferences. This issue is avoided in the validation and test sets because we re-annotated the answer spans. Furthermore, some of the generated options for the validation and test sets might also be acceptable, even if GPT-4 generates a completely different sentence.

**Is the dataset self-contained, or does it link to or otherwise rely on external resources (e.g., websites, tweets, other datasets)?** If it links to or relies on external resources, a) are there guarantees that they will exist, and remain constant, over time; b) are there official

archival versions of the complete dataset (i.e., including the external resources as they existed at the time the dataset was created); c) are there any restrictions (e.g., licenses, fees) associated with any of the external resources that might apply to a future user? Please provide descriptions of all external resources and any restrictions associated with them, as well as links or other access points, as appropriate.

We repuire the user to download the raw video data from the source video dataset including ActivityNet [4] and QVHighlights [7]. For ActivityNet, user can fill the form and get the permission to download raw video at `http://activity-net.org/download.html`. For QVHighlights, user can download the raw video at `https://github.com/jayleicn/moment_detr` with the provided download link `https://nlp.cs.unc.edu/data/jielei/qvh/qvhilights_videos.tar.gz`. For question a, they both provide a steady maintenance source of data on web. For question b, yes. For question c, please follow the licenses published by the source datasets. For the data, we follow QVHighlights and released inder CC BY-NC-SA 4.0 license. For code, we release under MIT license.

**Does the dataset contain data that might be considered confidential (e.g., data that is protected by legal privilege or by doctor-patient confidentiality, data that includes the content of individuals non-public communications)?** If so, please provide a description.

No.

**Does the dataset contain data that, if viewed directly, might be offensive, insulting, threatening, or might otherwise cause anxiety?** If so, please describe why.

No.

**Does the dataset relate to people?** If not, you may skip the remaining questions in this section.

Yes, some of the videos contain human activities.

**Does the dataset identify any subpopulations (e.g., by age, gender)?** If so, please describe how these subpopulations are identified and provide a description of their respective distributions within the dataset.

No.

**Is it possible to identify individuals (i.e., one or more natural persons), either directly or indirectly (i.e., in combination with other data) from the dataset?** If so, please describe how.

We only release question-answer data and corresponding time span annotations which do not contain any identifiable information. Each instance links to a source video. However, user and find people in the source video. These people in the video were de-identified according to the source video dataset.

**Does the dataset contain data that might be considered sensitive in any way (e.g., data that reveals racial or ethnic origins, sexual orientations, religious beliefs, political opinions or union memberships, or locations; financial or health data; biometric or genetic data; forms of government identification, such as social security numbers; criminal history)?** If so, please provide a description.

No.

**Any other comments?**

---



**Collection Process**



**How was the data associated with each instance acquired?** Was the data directly observable (e.g., raw text, movie ratings), reported by subjects (e.g., survey responses), or

indirectly inferred/derived from other data (e.g., part-of-speech tags, model-based guesses for age or language)? If data was reported by subjects or indirectly inferred/derived from other data, was the data validated/verified? If so, please describe how.

The video data, which is directly observable, was collected from the publicly accessible ActivityNet and QVHighlights datasets. In contrast, the text data was generated through the use of GPT-4. We prompt the LLM to find out the events pairs, generating question-answer pairs, filtering and scoring as depicted in **??**.

**What mechanisms or procedures were used to collect the data (e.g., hardware apparatus or sensor, manual human curation, software program, software API)?** How were these mechanisms or procedures validated?

We download the meta data and raw video from the two source video datasets at `http://activity-net.org/download.html` and `https://github.com/jayleicn/moment_detr`. We first filter out some samples by a rule-based method. We then use GPT-4 API to prompt the GPT-4 to find an event pair and categorize them into three target categories from the dense video captions or a video with description. After that, we prompt the GPT-4 to generate a question-answer pair and scoring them according to designed criteria. Additionally, we generate other 3 options for the assessment of multi-choice, and we re-annotated the answer time span by human annotators for validation and test sets.

**If the dataset is a sample from a larger set, what was the sampling strategy (e.g., deterministic, probabilistic with specific sampling probabilities)?**

We collect samples which are suitable for converting to reasoning across time QA. The rules we applied can are in Section 3.1.

**Who was involved in the data collection process (e.g., students, crowdworkers, contractors) and how were they compensated (e.g., how much were crowdworkers paid)?**

All participants are volunteering laboratory members as mentioned in the Acknowledgement. Due to the limited research budget, we usually conduct human verification or preference study within our research group. The participants in this project in return get help from us for their research need on human study.

**Over what timeframe was the data collected? Does this timeframe match the creation timeframe of the data associated with the instances (e.g., recent crawl of old news articles)?** If not, please describe the timeframe in which the data associated with the instances was created.

Generating QA data using the OpenAI GPT-4 API takes approximately five days. For human curation, we focused on the validation and test sets, comprising a total of 3,064 samples. Estimating that it takes about one hour to curate 60 samples, the entire human curation process requires roughly 50 hours.

**Were any ethical review processes conducted (e.g., by an institutional review board)?** If so, please provide a description of these review processes, including the outcomes, as well as a link or other access point to any supporting documentation.

No.

**Does the dataset relate to people?** If not, you may skip the remaining questions in this section.

Yes.

**Did you collect the data from the individuals in question directly, or obtain it via third parties or other sources (e.g., websites)?**

For raw video data, no, we collect data from existing video datasets. For QA data, we collect validation and test sets from human annotators directly.

**Were the individuals in question notified about the data collection?** If so, please describe (or show with screenshots or other information) how notice was provided, and provide a link or other access point to, or otherwise reproduce, the exact language of the notification itself.

For human annotators, they were notified about the data collection. We provide a gradio GUI for them for data labeling.

**Did the individuals in question consent to the collection and use of their data?** If so, please describe (or show with screenshots or other information) how consent was requested and provided, and provide a link or other access point to, or otherwise reproduce, the exact language to which the individuals consented.

Yes, all the annotators are volunteers.

**If consent was obtained, were the consenting individuals provided with a mechanism to revoke their consent in the future or for certain uses?** If so, please provide a description, as well as a link or other access point to the mechanism (if appropriate).

We didn't collect personal information in the annotation. We didn't provide a mechanism to revoke their consent in the future or for certain uses.

**Has an analysis of the potential impact of the dataset and its use on data subjects (e.g., a data protection impact analysis) been conducted?** If so, please provide a description of this analysis, including the outcomes, as well as a link or other access point to any supporting documentation.

Yes, we provide the discussion in Section Social impacts in supplementary.

**Any other comments?**

---

| Preprocessing/cleaning/labeling |
| :---: |

**Was any preprocessing/cleaning/labeling of the data done (e.g., discretization or bucketing, tokenization, part-of-speech tagging, SIFT feature extraction, removal of instances, processing of missing values)?** If so, please provide a description. If not, you may skip the remainder of the questions in this section.

Yes, preprocessing/cleaning/labeling of the data was done. The following steps were performed. For QVHighlights, videos where the answer span encompasses the entire duration were eliminated, and Samples with routine and repeated behaviors were excluded. For ActivityNet, samples with event duration exceeding 80% of the total video length were discarded and samples with cumulative segment duration less than 60% of the videos total length were excluded. Also, we cluster and select dense video captions from ActivityNet by the following steps:

1. Event intervals were clustered with a 10-second threshold.
2. Intervals separated by gaps exceeding this threshold were segmented into distinct groups.
3. The group with the highest event count was selected for generating question-answer pairs.

**Was the raw data saved in addition to the preprocessed/cleaned/labeled data (e.g., to support unanticipated future uses)?** If so, please provide a link or other access point to the raw data.

Yes, everyone can access the original data from the source datasets.

**Is the software used to preprocess/clean/label the instances available?** If so, please provide a link or other access point.

We provide the preprocessing code in our GitHub `https://github.com/ReXTime/ReXTime`.

**Any other comments?**

---

**Uses**

---

**Has the dataset been used for any tasks already?** If so, please provide a description.

At the time of publication, only the original paper have used this dataset.

**Is there a repository that links to any or all papers or systems that use the dataset?**
If so, please provide a link or other access point.

No.

**What (other) tasks could the dataset be used for?**

The dataset could be used for video understanding tasks such as moment localization, multi-choice VQA, generative evaluation.

**Is there anything about the composition of the dataset or the way it was collected and preprocessed/cleaned/labeled that might impact future uses?** For example, is there anything that a future user might need to know to avoid uses that could result in unfair treatment of individuals or groups (e.g., stereotyping, quality of service issues) or other undesirable harms (e.g., financial harms, legal risks) If so, please provide a description. Is there anything a future user could do to mitigate these undesirable harms?

No.

**Are there tasks for which the dataset should not be used?** If so, please provide a description.

No.

**Any other comments?**

---

**Distribution**

---

**Will the dataset be distributed to third parties outside of the entity (e.g., company, institution, organization) on behalf of which the dataset was created?** If so, please provide a description.

The dataset will be made publicly available and can be used for only research or non-commercial purpose.

**How will the dataset will be distributed (e.g., tarball on website, API, GitHub)** Does the dataset have a digital object identifier (DOI)?

Yes, we provide a GitHub project page https://rextime.github.io/ for all the link to our dataset. we provide a GitHub link for the code and download instruction https://github.com/ReXTime/ReXTime. We also hold a Huggingface dataset for the dataset maintenance https://huggingface.co/datasets/ReXTime/ReXTime. Users can find Croissant meta data and DOI on the Huggingface dataset page.

**When will the dataset be distributed?**

The full dataset will be available upon the acceptance of the paper before the camera-ready deadline.

**Will the dataset be distributed under a copyright or other intellectual property (IP) license, and/or under applicable terms of use (ToU)?** If so, please describe this license

and/or ToU, and provide a link or other access point to, or otherwise reproduce, any relevant licensing terms or ToU, as well as any fees associated with these restrictions.

Yes, we release our collected data under CC BY-NC-SA 4.0 license following QVHighlights. We release our code under MIT license.

**Have any third parties imposed IP-based or other restrictions on the data associated with the instances?** If so, please describe these restrictions, and provide a link or other access point to, or otherwise reproduce, any relevant licensing terms, as well as any fees associated with these restrictions.

No.

**Do any export controls or other regulatory restrictions apply to the dataset or to individual instances?** If so, please describe these restrictions, and provide a link or other access point to, or otherwise reproduce, any supporting documentation.

No.

**Any other comments?**

| Maintenance |
| --- |

**Who will be supporting/hosting/maintaining the dataset?**

National Taiwan University, Graduate Institute of Communication Engineering, Vision and Learning Laboratory.

**How can the owner/curator/manager of the dataset be contacted (e.g., email address)?**

First author email: r12942106@ntu.edu.tw

**Is there an erratum?** If so, please provide a link or other access point.

No, but we will keep updating on our GitHub and web page if necessary.

**Will the dataset be updated (e.g., to correct labeling errors, add new instances, delete instances)?** If so, please describe how often, by whom, and how updates will be communicated to users (e.g., mailing list, GitHub)?

Yes, we will update and announce on the web page and GitHub if necessary.

**If the dataset relates to people, are there applicable limits on the retention of the data associated with the instances (e.g., were individuals in question told that their data would be retained for a fixed period of time and then deleted)?** If so, please describe these limits and explain how they will be enforced.

We didn't collect personal information in the data collecting

**Will older versions of the dataset continue to be supported/hosted/maintained?** If so, please describe how. If not, please describe how its obsolescence will be communicated to users.

Yes, all users can access the old version of data on ours GitHub repo and Huggingface dataset.

**If others want to extend/augment/build on/contribute to the dataset, is there a mechanism for them to do so?** If so, please provide a description. Will these contributions be validated/verified? If so, please describe how. If not, why not? Is there a process for communicating/distributing these contributions to other users? If so, please provide a description.

Yes, contributions will be made possible via GitHub or email, submitted as pull requests to the relevant GitHub repository. We will provide the annotating and reviewing code to whom want to validate/verify the modified results.

**Any other comments?**