# OpenReview forum: "ReXTime: A Benchmark Suite for Reasoning-Across-Time in Videos"
_NeurIPS.cc/2024/Datasets_and_Benchmarks_Track — NeurIPS 2024 Track Datasets and Benchmarks Poster_

### Official Review · Reviewer_ySRg · 2024-07-21
**Review for 1551-ReXTime**

**Rating:** 6
**Confidence:** 5
**Correctness:** Yes.
**Clarity:** Yes

**Review:**

Pros:

1.	The benchmark is generally developed in a sound way.
2.	The baselines are strong and provide helpful insights.


Cons:

1.	The biggest concern is that I did not find fundamental difference between this benchmark and NExT-GQA. Also, the annotated number of val/test samples are even smaller (less than half). In addition, NExT-GQA contains a large portion of 'Causal' questions , but this paper ignores them and only compares with the temporal subset. This makes the comparisons (Tab.1 and Tab. 5) not convencing.
2.	Since the QAs are generated from video captions, it would be interesting to see the QA performance with either the original or the SOTA captions.
3.	While the benchmark provides clear question types, the analyses are not tailored to this design. It would be better to analyse the model performances across different type of questions. Moreover, the connection between the “means-to-an-end” questions and video understanding is weak. It sounds more like common sense.
4.	While the task is set to be a multi-choice QA, the paper (even supplementary) does not provide any examples to see the QA quality. According to the prompts in supplementary, it seems that the correct answer is always at the first option. I am thus concerned about the dataset biases. I suggest the author to conduct a “BlindQA” study to check how many questions can be answered by pure language.

**Strengths:**

see pros..

**Additional Feedback:**

no

**Documentation:**

How are the human annotators employed and paid?

**Limitations:**

See cons.

**Opportunities For Improvement:**

Need more detailed comparison with NExT-GQA.
Need more experiments and analyses.

**Relation To Prior Work:**

No, see my cons

**Summary And Contributions:**

This paper builds a ReXTime benchmark to challenge Video-LLMs for reasoning across time in videos. The task format is set to be multi-choice VideoQA, where both the questions and candidate answers are generated by prompting LLMs with caption inputs from both ActivityNet-captions and QVHighlight. The authors provide the results of advanced MLLMs on this task and show their limitations in performing reasoning across time.

---

> ### Author Rebuttal · Authors · 2024-08-17
>
> **Q1. Fundamental difference between ReXTime and NExT-GQA. Smaller val/test sets for ReXTime (vs. NExT-GQA). And, a large number of “Causal” questions from NExT-GQA are disregarded, and only comparisons with the temporal subset (i.e., comparisons not convincing).**
>
> Our ReXTime is designed to enable VLM with reasoning across time abilities, with data significantly enriched with cross-time reasoning features (not the focus of NExT-GQA).
>
> NExT-GQA contains 8911 QA pairs with time spans of either temporal (i.e., Qs with before/after/while) and causal (i.e., Qs with why or how) categories. There are 3358 val data (Temporal: 1403, Causal: 1995) and 5553 test data (Temporal: 2301, Causal: 3252). However, a large portion of their QAs are from the same time span (i.e., no cross-time reasoning), or their QAs are simply describing the associated temporal relationship (i.e., no reasoning information). Its paper lists selected examples as follows:
>
>     a. Why does the woman sway her body? -> Dancing while playing ukulele.
>     b. Why did the boy walk away from the woman? -> To take a paper
>     c. What did the boy do after he put the paper on his body? -> Use it as a costume.
>     d. Why are there two men standing at the center island and holding their camera? -> Recording.
>     e. Why does the man wearing cap backwards have to squat down when the car approaches him? -> Take photo.
>     f. Why are there fences separating the audiences -> Mark out area to drive
> We note that, ( c) and (e) are from the temporal category, and only ( c) requires a VLM to perform reasoning across time. The rest of the examples are from the causal category, but only (b) involves QA across time segments (i.e., Q and A of (a), (d), and (f) are all presented in the same time span).
>
> We have Table 5 present QA-mIoU scores, comparing the QA time span differences between NExT-GQA and ours. Our ReXTime has a low QA-mIoU of 15.5, while that of NExT-GQA is 66.0, confirming NExT-GQA with a large overlap between their QA pairs. Moreover, we report the QA-mIOU of NExT-GQA in the causal category, and the resulting score is extremely high as 85.63. This is the reason why we choose to disregard the data of this category. After such dataset processing, the size of NExT-GQA is 1,403/2,301 for val/test sets, and that of ours is 921/2,143 for val/tests. Note that we also collect a fully supervised training set of size 9,695, while the training data of NExT-GQA does not contain any time span annotation.
>
>
> **Q2. QAs are generated from the captions of existing benchmarks. Would be interesting to see QA performance with either the original or those produced by SOTA captioning models.**
>
> While we are not able to conduct the suggested experiments due to the limited rebuttal period, we would like to clarify why QA pairs generated by SOTA captioning models instead of the ground truth ones would not be expected to provide comparable VQA performances.
>
> Take VQA data collected from ActivityNet [7] as an example, we select the event pairs and their ground truth captions to prompt GPT4, generating QAs via in-context learning (see Sect. 3.2). While one can alternatively use captioning models like GPT-4v to generate captions, followed by the above procedure to generate QAs, language models are known to suffer from hallucination [Bai et al. (2024)]. As a result, their QA pairs would not accurately describe the semantics of the input video data, resulting in degraded VQA performance.
>
> **Q3. Performances of the three collected question types. And, the connection between the “Means-to-an-End” questions and video understanding seems weak (common sense?).**
>
> We now report the VQA performances of three different categories (i.e., *Sequential*, *Cause-Effect*, and *Means-to-an-End*) using API models. For GPT-4o, the accuracies on these three categories are 82%, 75%, and 66%; for Reka 69%, 59%, and 51% are optained. The performance of Sequential is the best among all. This is expected, since VQA data of this category simply describes the temporal relationship between an event pair. The latter two categories require the ability of reasoning across time, and thus degraded performances are observed.
>
> This is the reason why we build the dataset of ReXTime. Take VTimeLLM [17] as an example, after finetuning on our dataset, the VQA performances are improved from 37.37% to 58.59% and 35.4% to 62.69% on cause-effect and means-to-an-end VQA, respectively.
>
> To show that solving “means-to-an-end” VQA is not trivial (i.e., not simply a video understanding or QA task), we conduct an additional experiment on such data using GPT-4o with BlindQA (i.e., only Q as the input but not video) and VQA. We have the accuracies of 33% and 66%, respectively. This confirms that it is desirable for a VLM model which takes both video data and the question to perform temporal reasoning, so that the “*Means-to-an-End*” VQA task can be tackled accordingly.
>
> **Q4. With multi-choice QA, need examples to assess the QA quality. According to supplementary, it seems that the correct answer is always at the first option (possible dataset biases?). Suggest to conduct a “BlindQA” study.**
>
> While we list the correct answer as the first option (as shown in the supplementary), random shuffling is performed before conducting all the experiments in our work (for both API and open-source models). Thus, the comparisons are fair.
>
> In order to assess the QA quality, we follow the reviewer's suggestion and compare VQA with with BlindQA. Using GPT-4o, the performances of VQA and BlindQ are 73.67% and 29.67%, respectively. Thus, it is clear that such an API model is not able to produce satisfactory performance via BlindQA. This confirms the QA quality of our benchmark, suggesting that solving a temporal reasoning task is not trivial.

---

> > ### Comment · Reviewer_ySRg · 2024-08-19
> >
> > I appreciate the authors' careful responses to my questions.  The rebuttal has addressed well Q3 and Q4. However, I do not quite agree with the answers for Q1 and Q2.
> >
> > First for Q1, although NExT-GQA annotate a single time span that covers both question and answer time periods (usually question and answer are close to each other though not in the same time period), it does not mean that answering the question does not need to reason across time. The challenge is not related to evaluation labels. I am not sure how the authors obtain the IoU results. Do you re-annotate for the QAs or directly use the time span annotations of NExT-GQA for comparison?
> >
> > Second for Q2, the reason that I want to see this result is that I am expecting some new technique challenges to be a new benchmark, other than what can be solved in dense video captioning.
> >
> > Given other reviewers' comments and the authors have addressed part of my questions, I increase my score to 6 (above acceptance) and suggest the authors to add more related analyses about my point 1 and 2 in the final version.

---

> > > ### Author Rebuttal · Authors · 2024-08-20
> > >
> > > We sincerely thank the reviewer for providing insightful suggestions. We understand and agree that, while only a single time span is annotated for NExT-GQA, it does not mean that the trained models cannot perform reasoning across time. In our work, we follow the evaluation setting of EgoSchema [26]. That is, video data are sampled from each dataset, with the total sampled video length as 2 hours. With such sampled video data, time spans for each QA are annotated for calculating the corresponding IoU scores. So yes, as noted by the reviewer, we need to re-annotate the QA time spans of each dataset (including NExT-GQA) for comparison purposes.
> > >
> > > We understand the reviewer's concern on the use of dense video captions for solving VQA. It is true that, if dense video captions can be provided to faithfully describe the video, many vision-language tasks can be solved accordingly (e.g., VQA, video-text retrieval, moment retrieval, etc.). However, generating high-quality dense video caption is still an ongoing research topic [16, 31, 32], which requires the model to describe not only fine-grained but also temporally related captions. Nevertheless, this is indeed a very interesting point and worth further study. We will include this among our ongoing research directions.
> > >
> > > Once again, we sincerely thank the reviewer for the constructive feedback, and we commit to incorporating your suggestions into the revised version.

---

### Official Review · Reviewer_Sauq · 2024-07-23
**Official Comments**

**Rating:** 7
**Confidence:** 4
**Correctness:** Yes
**Clarity:** Yes

**Review:**

This paper proposes REXTIME, a benchmark designed to rigorously test AI models’ ability to perform temporal reasoning within video events. A common weakness shared by frontier MLLMs is discovered, i.e., they reason poorly when question 58 and answer span do not overlap. The LLM-assisted data pipeline generates high quality samples with reduced human intervention and points out a promising way for future model to equip the capability.

The dataset is constructed in a sound way. This paper is well-written. The supplementary material gives adequate supplements about the implementation and dataset details.

**Strengths:**

This paper proposes REXTIME, a comprehensive and reliable benchmark for multimodal AI, emphasizing reasoning-across-time and visual event localization in videos, with minimal human labor. The evaluation results demonstrate that even frontier MLLMs found REXTIME difficult and fall far behind human-level performance. The automatically constructed training dataset further points out a promising way for future model to equip the capability.

**Additional Feedback:**

None

**Documentation:**

Yes

**Limitations:**

See "Opportunities For Improvement"

**Opportunities For Improvement:**

(1) The motivation to compare academic and open source models was not clearly described.
(2) The difference between Intentionality and Purpose is confusing. Why “Adding ingredients into a cup. / Putting a drink on the table.”
should score high in Intentionality but low in Purpose?
(3) More examples of event pairs can make the three categories of question more understandable. The examples in Fig. 3 are clear and well understandable. But, I think there are many questions which can not be definitely classified as some category.

**Relation To Prior Work:**

Yes

**Summary And Contributions:**

This paper introduces a benchmark designed to rigorously test AI models’ ability to perform temporal reasoning within video events. The authors develop an automated pipeline for generating temporal reasoning question-answer pairs, significantly reducing the need for labor-intensive manual annotations.  Some academic and open source models are evaluated on the benchmark.

---

> ### Author Rebuttal · Authors · 2024-08-17
>
> **Q1. Why compare API with open-source models?**
>
> (1) To show the performance gap:
>
> API models such as GPT4o and Gemini are known for promising capabilities in handling multimodal learning tasks. Since their models are not publicly available, and one cannot access their training data either, open-source models provided by the research communities generally are not able to achieve comparable capabilities. This can be seen from paper Section 4.3, in which we compare the performance on our benchmark using such models.
>
> (2) To support the need for collecting data enriched with temporal reasoning information, allowing research communities to finetune open-source models
>
> Utilizing the benchmarks of ActivityNet [7] and QVHighlights [21], we propose a data processing pipeline to collect the dataset of ReXTime, which consists of VQA data in three different categories (i.e., *Sequential*, *Cause-Effect*, and *Means-to-an-End*). Using this proposed dataset to finetune open-source models, we observe significantly improved performances on the tasks of VQA and event localization, as presented Tables 3 and 4 in the main paper. Take the SOTA model of VTimeLLM [17] as an example, its performance after finetuning on our benchmark is observed to be comparable to that of the API model of Reka [3] (see table below). Thus, the use of our collected dataset for improving temporal reasoning capabilities of vision-language models can be successfully verified.
>
> ||Accuracy (%)|Accuracy(%)@IoU≥0.5|
> |-|-|-|
> | SOTA academic models before fine-tuned (VTimeLLM[17]) | 36.25 | - |
> | SOTA academic models after fine-tuned (VTimeLLM) | 58.15 (+21.90) | 18.30 |
> | Worst performing proprietary model (Reka-Core [3]) | 59.67 | 17.00 |
>
> **Q2. The difference between Intentionality and Purpose is confusing. Why “Adding ingredients into a cup. / Putting a drink on the table.” should score high in Intentionality but low in Purpose?**
>
> *Intentionality* and *Purpose*, together with *Directness* and *Necessity*, are the four criteria for us to categorize the collected VQA data (i.e., *Sequential*, *Cause-Effect* and *Means-to-an-end*). Please refer to Fig. 3 (or L114-119) for the definitions of the three data categories.
>
> The definitions of the above four criteria are presented In L134-150. *Directness* assesses the event causality between the two events (e.g., girl falls vs. cries, or band playing instruments vs. man taking pictures of the band). *Necessity* measures whether a successor event is the direct outcome of a predecessor event (e.g., girl falls vs. cries, or switch the light vs. light is on). *Intentionality* assesses whether a predecessor event is actively carried out with deliberate intention. It can be viewed as proactiveness disregard of a successor event. *Purpose* determines whether a successor event is the goal of a predecessor event (e.g., open bottle vs. drink water). Each criterion is scored by GPT-4, ranging from 0 (lowest) to 3 (highest). How each of the above VQA categories is associated with the four criteria is detailed in Sec. 3.2.
>
> For the example of the event pair “adding ingredients into a cup vs. putting drink on the table”, its *Intentionality* score is high since “adding ingredients into a cup” is an event actively performed. However, its *Purpose* score is low, since “putting drink on the table” is not the goal of “adding ingredients into a cup”.
>
> **Q3. More examples of event pairs for the collected three VQA categories. What if questions that cannot be classified into one of these three categories?**
>
> Additional VQA examples of each category are provided in Fig. 6 of supplementary (i.e., two more example VQA pairs provided for each VQA category). For each category in this figure, one example has the question event occurring before the answer event, while the other presents the opposite case.
>
> We understand the concern about the categorization of VQA data. As discussed in L152-153, VQA data not associated with *Cause-Effect* or *Means-to-an-End* will be categorized as *Sequential*. This is due to the fact that the event pair extracted from the same video must exist in temporal causality. Thus, when constructing our dataset, we first assess whether the event pair meets the criteria of *Cause-Effect* or *Means-to-an-End* categories; if not, it will be viewed as the sequential type.
>
> We fully agree that it is possible to define a novel category for describing temporal-reasoning data. We can either use the above existing four criteria to perform such categorization, or unique criteria can be proposed to categorize such VQA data. Either way still aligns the goal of our work, which presents an automated data processing pipeline and thus constructs VQA data with sufficient temporal reasoning features.

---

### Official Review · Reviewer_fKEv · 2024-08-08

**Rating:** 6
**Confidence:** 4
**Correctness:** Mostly correct, please refer my comme…
**Clarity:** yes.

**Review:**

read other sections for review.

**Strengths:**

1. The paper addresses a significant research question to improve temporal understanding of visual-language models, and highlight how much capability they have in looking at different parts of the video to answer the question at hand.

2. The dataset curation pipeline is cost-effective as it is a human-machine hybird, although I wonder how did the authors come up with the 300 vs 135 calculation in L34.

3. To understand the model performance beyond direct text-based reasoning, the authors also include moment extraction in time as an auxiliary task, where the model also has to predict the start and end times of the relevant answer snippet.

4. Reasonable evaluation of both large-scale MLLM models (called frontier models) as well as smaller-scale models (which are called academic models) showing their limitations in achieving good performance on this task.

**Additional Feedback:**

refer my questions/suggestions above.

**Documentation:**

yes.

**Ethics:**

no.

**Limitations:**

The authors have included a limitation section in the supplementary.

**Opportunities For Improvement:**

1. I wonder the possibility of more than one answer for the questions posed, especially in the means-to-end kind of siatuation. For example, the question `how to make a dish` is quite generic, and any of the previous steps (cut vegetables, chop tomatoes, boil water etc.) are all valid answers. Therefore, how do the authors delineate these kind of mistakes from the mistakes committed due to poor reasoning-in-time? Similarly, "What do S do before watching TV" rarely has a single unique answer.

2.  I really think the performance of GPT-4o on VQA accuracy is actually pretty strong (73%) - did the authors try some prompt-tuning strategies or in-context prompting? For the moment localization, it is expected that the frontier models have poor performance since they are not designed to accurately localize snippets in videos (as opposed to describe the action in plain text). Based on this, I wonder how much of the poor performance of these models is due to difficulty in the semantics of the task as opposed to reasoning-in-time.

3. It would be interesting to see a graph showing the effect of synthetic training data vs. the performance boost obtained, like in Table 4. Of course, this is unreasonable for a rebuttal timeline, but hopefully the authors could keep this in mind for the next version of the paper.

4. L123 and Fig 2: It is not clear if the extension is with respect to 10 frames or 10 seconds?

5. I am curious to know why is there a peak around 3-min mark in Fig 4? Also, the caption for Fig 4 seems to be mismatching with the figure order.

**Relation To Prior Work:**

yes.

**Summary And Contributions:**

The paper presents a new benchmark for evaluating reasoning-in-time of MLLM models, or really any video-language models in general. They argue that current benchmarks do not truly evaluate accurate reasoning in time within a video, so they curate a Q&A task which mandates the models to refer to distant frames in time, often non-intersecting with the reference clip. They devise a LLM-assisted dataset curation pipeline resulting in human filtered validation and test sets for evaluation, as well as unfiltered train sets, which is shown to improve performance when used for fine-tuning.

---

> ### Author Rebuttal · Authors · 2024-08-17
>
> **Q1. What if more than one correct answer? How to define a before/after event?**
>
> We follow [26, 38] and consider MCQs (Multiple Choice Questions). While MSQs (Multiple Select Questions) would be more general, to the best of our knowledge, there is no MSQ dataset available for VQA. Since our focus is to build a VQA benchmark for improving VLM’s temporal-reasoning ability, only MCQ is considered in our work.
>
> We now explain how we alleviate possible MSQs when collecting our ReXTime. When generating QA options, we start from the dense captions of the extracted event pairs. We take such captions as the input prompts to GPT4, which is applied to output one correct (from the input caption) and three unrelated options. However, we agree that it is possible for the generated options to include correct answers (presented in the input video). This would occur when the events associated with such answers are not fully described in the original captions. Nevertheless, it is worth repeating that our focus is to collect a dataset of performing cross-time QA reasoning. With open-source models finetuned with such temporal-reasoning information, improved performance can be obtained (e.g., grounding VQA as shown in Table 4).
>
> We agree with the reviewer that there might be multiple events for the before/after data category. Following the setting of NExT-QA[37]. We only focus on neighboring and consecutive events (i.e., closest time segments) in each video, as depicted in L153. This allows us to focus on constructing VQA data with reasoning-in-time properties. We understand that it is possible and even more desirable to consider MSQs instead of MCQs. However, to the best of our knowledge, there is no MSQ benchmark available for existing VQA works. We will list this in our future research directions, and we sincerely thank the reviewer again for the insightful suggestions.
>
> **Q2. Performance of GPT-4o on VQA is very strong (73%). Any prompt-tuning strategies or in-context prompting utilized? For moment localization, how much of the poor performance is due to difficulty in semantics instead of reasoning-across-time?**
>
> Yes, with the prompting strategy deployed, GPT-4o demonstrates impressive capabilities in temporal reasoning. We note that the same prompting strategy is applied for all API and open-source LLM-based models. While presented in Sect. C.2 of supplementary, our steps are described as follows:
>
>     1. Watch and briefly summarize the video.
>     2. Given 50 video frames and a question, choose the most correct answer from the options.
>     3. Find the time span in seconds that support your answer. The time span should be consistent with the option you choose.
> To further assess the difficulties of semantics misalignment vs. reasoning-in-time, we take GPT-4o and perform the following experiment. In Task #1, we report its performance of moment retrieval using the answer as the query. On the other hand, we have Task #2 consider the task fo grounding VQA, which take the video and the associated question as inputs, with the answer including the corresponding time span. The goal of Task #1 is to see how semantics alignment poses challenges in moment retrieval, while Task #2 is to assess how the ability of reasoning-in-time affects the localization of the corresponding time span. From the table below, we see that performances of Task #2 are inferior to those of Task #1, confirming that reasoning-in-time is indeed a more challenging task to tackle.
>
> |GPT-4o|mIoU|R@1(IoU=0.3)|R@1(IoU0.5)|
> |-|-|-|-|
> |Task #1: Moment retrieval w/ answer as query directly|41.56|57.00|40.00|
> |Task #2: Grounding VQA (i.e., answer and its time span as the outputs)|38.61|47.00|35.00|
>
> **Q3. Would be interesting to see a graph showing the effects of synthetic training vs. performance boosts like Table 4 (limited rebuttal timeline though).**
>
> While we haven't presented this data in a graph in the current version, we have provided relevant information in our existing tables. Specifically, Table 3 shows the zero-shot performance (i.e., before finetuning) on our test set, and Table 4 presents the performance of fine-tuned on our generated training data. To provide a clearer comparison, we have compiled the following table, showing the performance boost is observed for all open-source models after fine-tuning.
>
> ||Moment Retrieval|||VQA|
> |-|-|-|-|-|
> ||mIoU|R@1(IoU=0.3)|R@1(IoU=0.5)|Acc(%)|
> |UniVTG [24] (zs)|28.17|41.34|26.88|-|
> |UniVTG (ft)|34.63(+6.46)|53.48(+12.14)|34.53(+7.65)|-|
> |CG-Detr [28] (zs)|23.87|31.31|16.67|-|
> |CG-Detr (ft)|26.53(+2.66)|39.71(+8.40)|22.73(+6.06)|-|
> |VTimeLLM [17] (zs)|20.14|28.84|17.41|36.16|
> |VTimeLLM (ft)|29.92(+9.78)|43.69(+14.85)|26.13(+8.72)|57.58(+21.42)|
> |TimeChat [33] (zs)|11.65|14.42|7.61|40.04|
> |TimeChat (ft)|26.29(+14.64)|40.13(+25.71)|21.42(+13.81)|49.46(+9.42)|
>
>
> **Q4. L123 and Fig 2: It is not clear if the extension is with respect to 10 frames or 10 seconds?**
>
> While collecting event pairs from QVHighlights[21] in our data generation pipeline, we extend the video frames from the existing annotated span (pivotal event) to acquire neighbor events information with VLMs. We extend 5 frames to the predecessor and successor frames of the pivotal event, respectively (i.e., a total of 10 frames). For QVHighlights, since the frame rate is 2 seconds per frame, the above extension would indicate a 10-second extension in both directions (i.e., a total of 20 seconds).
>
> **Q5. Why is there a peak around 3-min mark in Fig 4? The caption for Fig 4 is mismatching with the figure order.**
>
> For the validation and test sets, we collect videos and annotations from the ActivityNet[7] and QVHighlights[21] datasets. Video durations in the former are normally distributed, and while those from the latter are all 150 seconds. This is the reason why there is a peak around 3 mins. We also thank the reviewer for pointing out the caption mismatch, which will be corrected in our future version.

---

> ### Comment · Area_Chair_Sxe7 · 2024-09-01
> **Review/rebuttal discussion**
>
> Dear Reviewer fKEv,
>
> Please let us know if the rebuttal changes your opinion about the paper?
>
> Thanks

---

### Official Review · Reviewer_QTdk · 2024-08-13
**REXTIME dataset paper review**

**Rating:** 7
**Confidence:** 4

**Review:**

- This dataset can an important addition in the field of video understanding in general.
- The automated pipeline to create the dataset is thorough.
- The manually vetted validation and test set can provide a reliable benchmark to assess MLLMs performance on across-time reasoning.
- Analysis on proprietary MLLMs performance on the test dataset was able to point out these current advance models drawback and room for improvement.
- The training dataset helped to fine-tune open-source models and improved their performance on across-time reasoning.

**Strengths:**

Assessing and improving MLLMs performance in reasoning-across-time tasks is beneficial to develop overall vision language logical understanding of these models. Presently, there is a lack of dataset focusing on the said task, and in this context this dataset is important. The work covers many aspects to make the dataset diverse and complex, along with a detailed data statistics.

**Additional Feedback:**

No.

**Clarity:**

The paper is mostly well-written. I had some questions which I posed in the above sections. Improving those will make the work only better.

**Correctness:**

The paper appears to be mostly correct. The questions I have are noted in the "Opportunities For Improvement" section.

**Documentation:**

The paper is mostly well written.

**Ethics:**

No.

**Limitations:**

Please refer to the "Opportunities For Improvement" section.

**Opportunities For Improvement:**

- line 45, "we have address" -> "we have to address".
- How does the performance of the fine-tuned models are affected when the distance between question and answer time segments increases?
- To provide accurate classification, this QA dataset provides multi-choice. A more general structure in future will be even better.
- In related work, how event localization and moment retrieval are different?
- Line 130, "less similar pairs". Is there threshold to filter similar pairs?
- Only causality and intentionality are shown in Fig. 3. Including all four scoring will be more reflective of the discussion in page 5.
- What are the ranges of the four scores?
- How is it ensure that there is no data leakage across partitions?
- Line 187, "thresholds of 0.3 and 0.5 at various thresholds". What does it mean?
- Line 192-193, "For evaluating ...". This sentence is not clear.
- Page 9, Fig. 4 caption. The middle histogram and the right histogram should be the other way around.
- Line 252, "video duration". The complete video duration or event pairs individual or total duration?

**Relation To Prior Work:**

There are prior works / datasets present, which address the problem of reasoning across time in an implicit manner. That is, there are subset of datasets available, that can be used of such task, but not an explicit dataset with thorough analysis.

**Summary And Contributions:**

In this work, the authors focused on the task of reasoning across time in video. They noted the absence of proper test data to evaluate current Multimodal Large Language Models (MLLMs) ability in reasoning across time.
To this end they introduced the dataset REXTIME. Here, the authors proposed an automated pipeline for generating temporal reasoning question-answer pairs from captioned videos in QVHighlights and ActivityNet using Multimodal (GPT-4V) and simple (GPT-4) LLMs. They categorized the samples into three classes (sequential, cause-effect, and means-to-an-end) using four scoring criteria (directness, necessity, intentionality, and purpose). This was done to ensure diversity. The resultant dataset contains 9,695 training, 921 validation, and 2,143 test samples. The validation and test samples were manually vetted for accuracy and relevance.
Using 300 sampled test split, this work showed that while the proprietary models perform better than open-source models in across-time reasoning, they still fall behind human level logic. Though not manually verified, by using the training partition to fine-tune the open-source models, was abled to improve their performance on such tasks. The work also showed that, in terms of average certificate lengths (CL) and question-answer interaction of union (QA-IoU), the proposed dataset offers more complexity and diversity.

---

> ### Author Rebuttal · Authors · 2024-08-17
>
> **Q1. How is VQA performance affected when the distance between QA time segments increases?**
>
> Due to limited rebuttal time, we sample 50 videos with an average length of 2 min from our dataset for evaluation. With 3 seconds as the threshold, the number of videos with a time difference between QA segments less than 3 seconds is 38; on the other hand, 12 videos are with QA time difference above 3 seconds. Using pre-trained VTimeLLM [17], the VQA results using data with gap below and above 3 secs are 26.32% and 42.11%, respectively. After finetuning on our benchmark, the results are 33.33% and 83.33%, respectively. Thus, while the performance gain can be observed for both cases, a large improvement (+50%) can be seen for videos with larger QA time gaps. Therefore, using our collected dataset for finetuning is desirable for the task of temporal reasoning, especially when the distance between QA segments becomes larger.
>
> **Q2. This QA dataset provides multiple choices. What about a more general structure in the future?**
>
> We thank the reviewer for the helpful suggestion. Our VQA dataset consists of MCQs (Multiple Choice Questions), where out of given 4 options, only one is correct. Please see L159-161 on how we generate MCQ data from the benchmark datasets of ActivityNet [7] and QVHighlights [21]. It is possible to consider MSQs (Multiple Select Questions) in VQA, allowing multiple correct choices to be presented. However, to the best of our knowledge, there is no MSQ dataset available for assessing VQA performances. Nevertheless, considering MSQs would make the VQA problem even more practical yet challenging, and we will list this among our ongoing directions.
>
> **Q3. Difference between localization and moment retrieval**
>
> Given a video, moment retrieval [21, 24, 28] refers to the task in which the model takes a language query as the input, aiming at producing one or more segments from that video matching the input query. On the other hand, event localization [39] is a more general task, which is typically accompanied by relevant yet more complex tasks like grounding VQA. (Ground VQA requires the model to perform reasoning across time, while identifying keyframes associated with the predicted answer).
>
> **Q4. Line 130: Is there a threshold to filter similar pairs? What are the ranges of the four scores?**
>
> Yes, since we adopted a score from the range 0 to 3 (output by GPT4) to calculate similarity scores and filter event pairs with a threshold of 2. An event pair collected from the benchmark dataset is discarded if its similarity score >= 2. Similarity score is also a criterion like *Directness*, *Necessity*, *Intentionality* and *Purpose* as depicted in Sec. 3.2. We adopt *Similarity* scores to avoid selecting semantically identical events as candidate pairs. Similarly, the scores of the four criteria range from 0 to 3, depicted in paper line 151.
>
> **Q5. Only causality and intentionality are shown in Fig. 3. Including all four scoring will be more reflective.**
>
> We thank the reviewer for the suggestion. In our work, we construct and categorize event pairs into three VQA types of *Sequential*, *Cause-Effect* and *Means-to-an-End*, as depicted in Fig. 3. The four scoring criteria of *Directness*, *Necessity*, *Intentionality* and *Purpose*, are proposed to perform the above categorization process. As noted in L151-155, our data classification rules are: (1) If the sum of *Directness* and *Necessity* scores (Causality in Fig. 3) is below 4, they are in a simple sequential relation. (2) If the sum of *Intentionality* and *Purpose* (Intentionality in Fig. 3) is less than 5, they are classified as a *Cause-Effect* relation. (3) If neither of the above conditions is met, the events are in a *Means-to-an-End* relation. We agree that it would be clearer if we add such selection criteria into Fig. 3, and we will be happy to take the reviewer’s suggestion to update this figure in our future version.
>
> **Q6. No data leakage across partitions?**
>
> We thank the reviewer for pointing this out. Data leakage occurs when the model is exposed to information shared by both the training and validation or test data (e.g., different segments from the same video split into training and validation sets). Take ActivityNet [7] for example, we always split its data based on video ID (i.e., entire video sequence in either training or validation set), and thus there is no data leakage issue.
>
>
> **Q7. Line 187, "thresholds of 0.3 and 0.5 at various thresholds". What does it mean?**
>
> We thank the reviewer for giving us the opportunity to clarify this concern. Since we considered two different thresholds for valuation, the edited line will be “We evaluate the Recall@1 score using two Intersection over Union (IoU) thresholds of 0.3 and 0.5.”
>
> **Q8. line 192-193 is not clear.**
>
> In L192-193, we have “For evaluating MLLMs, we prompt the models to predict the time span and select the most likely options.” This is to describe the strategy applied to perform VQA inference. We let the language models (such as GPT-4o) choose the correct answer from the four options, and see if the selected one matches the correct answer. Please see below for the prompts used to make the above request:
>
>     1. Watch and briefly summarize the video.
>     2. Given 50 video frames and a question, choose the most correct answer from the options.
>     3. Find the time span in seconds that support your answer. The time span should be consistent with the option you choose.
>
> Please refer to Sect. C.2 in our supplementary for the detailed prompting process.
>
> **Q9. Page 9, Fig. 4 caption. The middle histogram and the right histogram should be the other way around.**
>
> Thanks for the careful inspection. We will fix the order of the captions.
>
> **Q10. Line 252:  The complete video duration or event pairs individual or total duration?**
>
> L252 describes the dataset statistics depicted Fig.4. Thus, “video duration” refers to the total length of a video.

---

> > ### Comment · Reviewer_QTdk · 2024-08-26
> > **Reply to authors' rebuttal**
> >
> > Thank you authors for the response. I am satisfied and I will keep my score.
> > Please include the clarifications and answers in the final version for completeness.

---

### Author Response · Authors · 2024-08-17
**A breif summarization.**

Dear AC,

We sincerely appreciate the valuable time and insightful feedback provided by reviewers. We are grateful for the opportunity to address the concerns raised by each reviewer, which fundamentally strengthened our work. The strengths pointed out by the reviewers include:

(**Presentation**) The paper addresses a significant research question to improve temporal understanding of visual-language models, and highlight how much capability they have in looking at different parts of the video to answer the question at hand. [Reviewer fKEv]

(**Design choice**) The automatically constructed training dataset further points out a promising way for future models to equip the capability. [Reviewer Sauq]

(**Experiment**) Reasonable evaluation of both large-scale MLLM models (called frontier models) as well as smaller-scale models (which are called academic models) showing their limitations in achieving good performance on this task. [Reviewer fKEv]

(**Contribution**) Assessing and improving MLLMs performance in reasoning-across-time tasks is beneficial to develop overall vision language logical understanding of these models. Presently, there is a lack of dataset focusing on the said task, and in this context this dataset is important. [Reviewer QTdk]

We would like to point out that particular concerns are raised, as listed below. Please refer to the responses in our rebuttal for further details.

**QA quality assessment & a study of BlindQA**  [Reviewers ySRg]

**More than Multiple-choice-questions** (MCQs) [Reviewers QTdk, fKEv]

**How different from NExT-GQA?** [Reviewer ySRg]

**Why compare API with open-source models?** [Reviewer Sauq]

We thank the reviewers again for the suggestions and the raised issues. Given the recognized strengths from four reviewers, together with additional experiments and clarification and discussion provided during rebuttal, we hope this work will be of great value to multi-modal understanding communities, and the reviewers will be able to provide proper evaluation in the final decision. Thank you again for your kind attention and assistance.

Sincerely,
Authors of submission # 1551

---

### Decision · Program_Chairs · 2024-09-26

**Decision:**

Accept (Poster)

**Comment:**

The paper proposes a new benchmark, ReXTime which focuses on temporal reasoning tasks (reasoning across time) for fine-grained video understanding. All four expert reviewers favor the paper because of the novel video understanding tasks brought by the benchmark, the dataset curation pipeline is cost-effective, and the usefulness of the the dataset. AC agrees that the dataset will be useful for future research in fine-grained video understanding, thus recommends to accept the paper.